# Scalable Attribute-Missing Graph Clustering via Neighborhood Differentiation

Yaowen Hu [* 1]   Wenxuan Tu [* 2]   Yue Liu [1]   Xinhang Wan [1]   Junyi Yan [1]   Taichun Zhou [1]   Xinwang Liu [† 1]

## Abstract

Deep graph clustering (DGC), which aims to unsupervisedly separate the nodes in an attribute graph into different clusters, has seen substantial potential in various industrial scenarios like community detection and recommendation. However, the real-world attribute graphs, e.g., social networks interactions, are usually large-scale and attribute-missing. To solve these two problems, we propose a novel DGC method termed **C**omplementary **M**ulti-**V**iew **N**eighborhood **D**ifferentiation (*CMV-ND*), which preprocesses graph structural information into multiple views in a complete but non-redundant manner. First, to ensure completeness of the structural information, we propose a recursive neighborhood search that recursively explores the local structure of the graph by completely expanding node neighborhoods across different hop distances. Second, to eliminate the redundancy between neighborhoods at different hops, we introduce a neighborhood differential strategy that ensures no overlapping nodes between the differential hop representations. Then, we construct $K + 1$ complementary views from the $K$ differential hop representations and the features of the target node. Last, we apply existing multi-view clustering or DGC methods to the views. Experimental results on six widely used graph datasets demonstrate that CMV-ND significantly improves the performance of various methods.

## 1. Introduction

Deep graph clustering (DGC) aims to partition the nodes of an attributed graph into distinct groups (Liu et al., 2022a; Gong et al., 2024; Guan et al., 2025; Liu et al., 2025; Yu et al., 2024b). Typically, DGC first embed nodes into a latent space before performing clustering. Existing DGC approaches commonly rely on contrastive learning (Wan et al., 2024b; Tu et al., 2024c; Yang et al., 2024; 2023b) or reconstruction-based models (Li et al., 2023), either maximizing intra-cluster similarity or reconstructing node features to enhance clustering performance. In recent years, DGC has achieved remarkable success in various real-world applications, including community detection, metagenomic binning, and recommendation systems (Xue et al., 2022).

Despite their demonstrated effectiveness, most DGC methods are evaluated under conditions that have limited practical relevance. On one hand, they are typically tested on small-scale datasets, whereas real-world industrial and commercial applications often involve graphs containing hundreds of thousands or even millions of nodes (Ding et al., 2019), making it infeasible to apply standard Graph Neural Networks (GNNs) (Yin et al., 2024; Ju et al., 2024) directly at such scales (Lim et al., 2021). On the other hand, the success of existing DGC approaches hinges on the assumption that all graph samples are complete, an assumption frequently violated in practice (Huo et al., 2023). In many real-world scenarios, data collection is subject to privacy policies, copyright restrictions, or equipment failures, resulting in partially or completely missing attributes that significantly degrade overall clustering performance.

Although recent efforts have addressed DGC methods for large-scale graphs and missing attributes separately, the more realistic scenario, where both challenges coexist, remains underexplored. Notably, these problems are not orthogonal: missing attributes in large-scale graphs often give rise to additional complexities. In particular, large-scale graphs tend to be more sparse. For example, the ogbn-papers100M dataset contains around $1.11 \times 10^8$ nodes and $1.6 \times 10^9$ edges, edge density of only about $2.6 \times 10^{-4}$. In such highly sparse graphs, existing attribute completion methods become weak at inferring missing attributes, as they primarily rely on graph structure. Moreover, to scale

---

[*]Equal contribution  [1]College of Computer Science and Technology, National University of Defense Technology, Changsha 410073, China  [2]School of Computer Science and Technology, Hainan University, Haikou 570228, China. Correspondence to: Xinwang Liu † <xinwangliu@nudt.edu.cn>.

*Proceedings of the 42ⁿᵈ International Conference on Machine Learning*, Vancouver, Canada. PMLR 267, 2025. Copyright 2025 by the author(s).

to large graphs, DGC models typically rely on subgraph sampling for mini-batch training. This strategy inevitably alters the original topology and further weakens the limited structural cues, leading to degraded clustering results under missing attributes.

As described above, the key to addressing attribute-missing large-scale DGC lies in preserving graph structural information as completely as possible. Inspired by the selective attention theory (Dayan et al., 2000) in cognitive psychology, where human memory filters out redundant information and retains only the most critical incremental data for memory and cognition, we propose a novel paradigm called **C**omplementary **M**ulti-**V**iew **N**eighborhood **D**ifferentiation (*CMV-ND*). The core idea is to preserve graph structural information in multiple views in a complete and non-redundant manner. Specifically, *CMV-ND* consists of the following steps. First, we perform recursive neighborhood search (RNS) on each node to capture neighborhood information at various hop distances. To eliminate redundancy, we introduce a neighborhood differential strategy (NDS) that ensures there are no overlapping nodes between the differential representations of each hop, thereby achieving information redundancy reduction. Finally, the resulting $K$ differential hop representations, along with the original node features, form $K + 1$ complementary views, which are then used for the clustering.

With the above designs, *CMV-ND* achieves the goal of preserving graph structural information in multiple views in a complete and non-redundant manner. Unlike the traditional "aggregate-encode-predict" pipeline of GNNs, *CMV-ND* does not leverage graph structure through propagation operations. Instead, it directly stores graph structural information in the views through differential neighborhoods, enabling it to process large-scale graph data without sampling. Additionally, the complementary multi-view differential hop representations generated by *CMV-ND* offer high flexibility. On one hand, they can be fused (Wan et al., 2022) and input into any existing graph clustering method. On the other hand, they can also be directly applied to various methods in the multi-view clustering (MVC) (Wan et al., 2024a;c; Yu et al., 2024a; 2023). The main contributions are summarized as follows.

- This is the first attempt to perform DGC on large-scale graphs with missing attributes. Unlike approaches that focus solely on either attribute-missing graphs or large-scale graph clustering, our method addresses a more realistic and challenging scenario.

- We propose a novel graph clustering paradigm, *CMV-ND*, which preemptively includes graph structural information in multiple views in a complete and non-redundant manner, aiming to address the challenge of attribute-missingness in large graphs.

- Since *CMV-ND* constructs multi-view representations of nodes within the graph, it naturally bridges the gap between graph clustering and MVC.

- We validate *CMV-ND* through experiments on six widely used graph datasets, evaluating its superiority, sensitivity, efficiency, robustness, and effectiveness. Even a simple concatenation of the views generated by *CMV-ND* followed by direct application of $K$-means achieves superior performance compared to most existing DGC methods.

## 2. Relate Work

In this section we focus on the two most relevant areas to this work—attribute-missing graph clustering and large-scale deep graph clustering. Other related areas including deep graph clustering, large-scale graph learning, and attribute-missing graph completion are discussed in Appendix C for brevity.

### 2.1. Large-Scale Deep Graph Clustering

Scalable GNNs specifically tailored for clustering tasks are still limited. This is primarily because clustering tasks require the model to estimate the entire sample distribution at once. When the node count reaches the order of hundreds of millions, this often leads to memory shortages or excessive runtime. Recently, only two papers have attempted to scale DGC to large-scale graphs. Scalable self-supervised graph clustering ($S^3$GC) (Devvrit et al., 2022) uses lightweight encoders and simple random walk-based samplers to ensure that the embedding of a node is close to its "nearby" nodes while being far from all other nodes. Although its effectiveness has been validated, this method separates representation learning from clustering optimization, leading to suboptimal overall performance. Dilation shrink network (Dink-Net) (Liu et al., 2023b) proposes a new scalable method that unifies embedding learning and clustering into an end-to-end framework, which not only scales to large graph data but also learns clustering-friendly representations. However, all of the above methods rely on sampling, and as mentioned earlier, structural information is critical for large-scale graph clustering under attribute-missing conditions. Therefore, due to the disruptive effects of sampling on graph structure, these methods cannot be directly applied to large graphs in such scenarios.

### 2.2. Attribute-Missing Graph Clustering

In the domain of attribute-missing graph clustering, one representative approach is the attribute-missing graph clustering network (AMGC) (Tu et al., 2024a). AMGC addresses the dual challenge of clustering and attribute imputation by adopting an iterative framework that alternates between

the two. Specifically, it leverages the current clustering assignments to identify clustering-enhanced nearest neighbors, which are then used to refine the imputed attributes. This feedback mechanism allows the model to progressively enhance the feature quality of attribute-missing nodes and align the latent representations with the underlying cluster structure.

By integrating clustering signals into the imputation process, AMGC achieves competitive performance. However, the framework relies heavily on a GNN encoder, which performs full-graph message propagation across iterations. This reliance introduces significant memory and computational overhead, as the entire graph must be loaded into memory and processed in a holistic fashion. As a consequence, AMGC exhibits poor scalability and cannot be directly applied to large-scale graphs with millions of nodes and edges, where full-batch training becomes infeasible.

## 3. Method

### 3.1. Notations and Problem Definition

**Basic Notations.** *Let $\mathcal{G} = \{\mathcal{V}, \mathcal{E}, \mathbf{H}\}$ denote an undirected graph, where $\mathcal{V} = \{v_n\}_{n=1}^N$ is the set of vertex with $N$ nodes, $\mathcal{E}$ is the set of edges, $\mathbf{H} \in \mathbb{R}^{N \times d}$ is the node attribute matrix where $d$ denotes the feature dimension of the node. $\mathbf{A} \in \mathbb{R}^{N \times N}$ is the adjacency matrix that represents the relationships between nodes. Specifically, $a_{ij} = 1$ if there is an edge between nodes $v_i$ and $v_j$, and $a_{ij} = 0$ otherwise.*

**Definition 1 (k-hop neighborhood).** *For a node $v$, its $k$-hop neighborhood, denoted by $\mathcal{N}^k(v)$, is defined as the set of nodes whose shortest path distance to $v$ is less than or equal to $k$. Formally, $\mathcal{N}^k(v) = \{u \in \mathcal{V} \mid \mathrm{dis}(v, u) \leq k\}$, where $\mathrm{dis}(v, u)$ denotes the length of the shortest path between nodes $v$ and $u$. Note that $\mathcal{N}^k(v)$ includes node $v$ itself, since $\mathrm{dis}(v, v) = 0$.*

**Definition 2 (k-differential hop neighborhood).** *The $k$-differential hop neighborhood of node $v$, denoted by $\mathcal{D}^k(v)$, refers to the set of nodes that are exactly $k$ hops away from $v$, excluding nodes that are closer. Formally, $\mathcal{D}^k(v) = \{u \in \mathcal{V} \mid \mathrm{dis}(v, u) = k\}$. This definition captures the "difference" between the $k$-hop neighborhood and the $(k-1)$-hop neighborhood: $\mathcal{D}^k(v) = \mathcal{N}^k(v) \setminus \mathcal{N}^{k-1}(v)$. For example, consider a graph with edges $(v, a), (v, b), (a, c)$. Then: $\mathcal{N}^1(v) = \{v, a, b\}$, $\mathcal{N}^2(v) = \{v, a, b, c\}$. The corresponding differential hop neighborhoods are: $\mathcal{D}^1(v) = \{a, b\}, \mathcal{D}^2(v) = \{c\}$.*

**Definition 3 (attribute-missing graph).** *An attribute-missing graph is a graph where certain nodes lack feature representations. The node set $\mathcal{V}$ is divided into two disjoint subsets: $\mathcal{V}^c$ and $\mathcal{V}^m$, i.e., $\mathcal{V}^c$ denotes the set of attribute-*

complete nodes and $\mathcal{V}^m$ denotes the set of attribute-missing nodes, where $\mathcal{V} = \mathcal{V}^c \cup \mathcal{V}^m$ and $\mathcal{V}^c \cap \mathcal{V}^m = \emptyset$. Let $N^c = |\mathcal{V}^c|$ and $N^m = |\mathcal{V}^m|$, then the total number of nodes is $N = N^c + N^m$.

### 3.2. Challenge Analyses

This section analyzes the challenges faced by DGC in the context of large-scale graphs with missing attributes. Since clustering is an unsupervised task, a graph can be fully described using two sources of information, i.e., the feature view and the structural view. Specifically, let $\widetilde{\mathcal{G}} = (\mathcal{V}^m, \mathcal{E})$ represent the graph, where $\mathcal{V}^m$ denotes the feature view and $\mathcal{E}$ represents the structural view. In the case of attribute-missing graphs, the feature view is inherently incomplete. As a result, effectively leveraging the structural view becomes critical for improving clustering performance.

However, existing large-scale graph clustering paradigms often end up further compromising the structural view. On one hand, sampling commonly used to handle large graphs, disrupt the graph structure. On the other hand, the message-passing mechanism inherently introduces redundancy in node representations, as information from neighboring nodes is repeatedly aggregated during the process.

### 3.3. Proposed Solution

Based on the analysis in the previous section, the key to addressing attribute-missing large-scale DGC lies in preserving graph structural information as completely as possible. To tackle this challenge, we propose a novel graph clustering paradigm called *CMV-ND*. Intuitively, *CMV-ND* stores graph structural information in multiple views in a complete and non-redundant manner through preprocessing. *CMV-ND* consists primarily of two components, i.e., RNS and NDS. We provide a detailed explanation of these components in the following sections, including an overview of the overall architecture shown in Figure 1.

#### 3.3.1. RECURSIVE NEIGHBORHOOD SEARCH

The RNS aims to explore the neighborhood structure of the graph by recursively expanding the set of nodes at increasing hop distances. This process can be viewed as a breadth-first traversal, where in each step the neighborhood is extended by adding nodes that are exactly one hop away from the current set.

The recursive expansion of the neighborhood is defined by setting $\mathcal{N}^0(v) = v$, i.e., the node itself, and recursively applying the following relation to determine $\mathcal{N}^k(v)$:

$$\mathcal{N}^{k+1}(v) = \mathcal{N}^k(v) + \bigcup_{u \in \mathcal{N}^k(v)} \mathcal{N}(u), \quad (1)$$

where $\mathcal{N}(u)$ denotes the set of neighbors of node $u$. At each

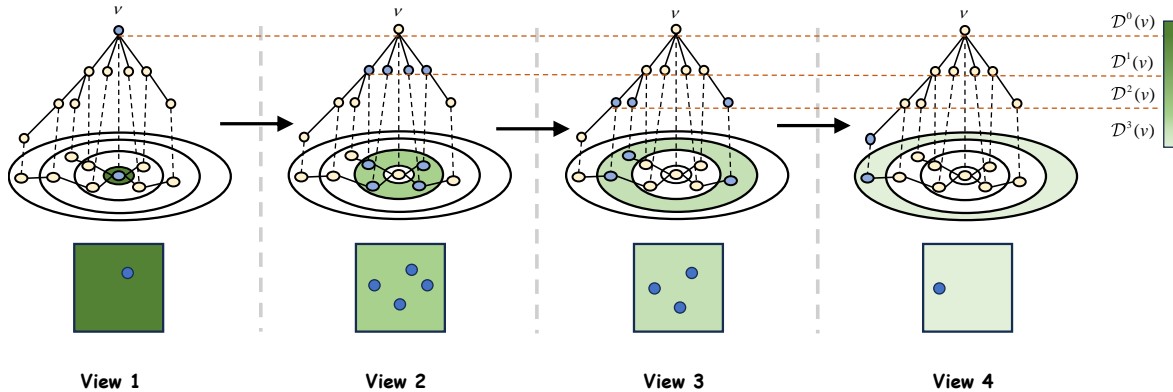

*Figure 1.* Overall workflow of the CMV-ND algorithm. The algorithm recursively searches the neighborhood of each node, treating each neighborhood as a distinct view. Each view contains complete nodes that do not overlap with other views.

step, new nodes are added to the neighborhood by exploring all neighbors of the previously discovered nodes, ensuring that nodes are processed layer by layer. The search halts once the $k$-hop neighborhood is fully determined, and no further neighbors are added.

### 3.3.2. NEIGHBORHOOD DIFFERENTIAL STRATEGY

Through the RNS, we obtain the $k$-hop neighborhoods of nodes in a non-propagative manner. However, we observe that the resulting neighborhoods are highly redundant, as the $(k + 1)$-hop neighborhood contains all nodes in the $k$-hop neighborhood. To address this redundancy, we propose a simple but effective NDS to separate distinct neighborhood information between different hops.

Formally, let the $k$-hop neighborhood of a node $v$ be denoted as $\mathcal{N}^k(v)$, which includes all nodes whose shortest path distance to $v$ is less than or equal to $k$. We define the $k$-differential hop neighborhood, $\mathcal{D}^k(v)$, as:

$$\mathcal{D}^k(v) = \mathcal{N}^k(v) \setminus \mathcal{N}^{k-1}(v), \qquad (2)$$

which contains only the nodes whose shortest path distance to $v$ is exactly $k$. This formulation isolates the structural information at each hop and avoids redundancy across different neighborhood levels.

The resulting differential hop neighborhoods, $\mathcal{D}^k(v)$, satisfy the definition provided in ***Definition 2***, where each differential hop neighborhood contains only nodes that are at an exact distance of $k$ from the node $v$, with no overlap with nodes from the previous hop. As shown in Algorithm 1, the full procedure of CMV-ND is presented.

### 3.3.3. CONSTRUCT MULTI-VIEW REPRESENTATION

To enable graph clustering, we propose to construct multi-view node representations based on the structural granularity of neighborhoods. The Recursive Neighborhood Search

(RNS) is employed to efficiently locate multi-hop neighbors, while the Neighborhood Differential Strategy (NDS) isolates the structural information at each $k$-differential hop. The representation of the $k$-differential hop neighborhood is computed by aggregating the features of all nodes in:

$$\mathbf{h}_v^k = \frac{1}{n_{vk}} \sum_{u \in \mathcal{D}^k(v)} \mathbf{h}_u, \qquad (3)$$

where $n_{vk}$ is the number of $k$-hop neighbors of $v$, $\mathbf{h}_u \in \mathbb{R}^d$ is the feature vector of the neighboring node $u$, and $\mathbf{h}_v^k \in \mathbb{R}^d$ is the aggregated representation of the $k$-hop neighborhood of node $v$.

Next, for each node $v$, we define its multi-view representation as $\mathbf{H}_v \in \mathbb{R}^{(k+1) \times d}$, which contains the following components:

$$\mathbf{H}_v = \left[ \mathbf{h}_v^0; \mathbf{h}_v^1; \dots; \mathbf{h}_v^k \right], \qquad (4)$$

where $\mathbf{h}_v^0$ denotes the original feature of node $v$, and $\mathbf{h}_v^k$ is the aggregated representation of the $k$-differential hop neighborhood. Therefore, the multi-view representation $\mathbf{H}_v$ captures both local and progressively expanded structural information, forming a $(k+1) \times d$ matrix.

To construct the graph-level multi-view representation, we stack the $(k + 1)$-view representations of all nodes along the view dimension. This results in a 3D tensor $\mathcal{H} \in \mathbb{R}^{(k+1) \times N \times d}$, where $N$ is the number of nodes and $d$ is the feature dimension. Formally, it is written as:

$$\mathcal{H} = [\mathbf{H}_1; \mathbf{H}_2; \dots; \mathbf{H}_N]. \qquad (5)$$

We describe how the above multi-view representations are utilized in Appendix B.

### 3.3.4. WHY CMV-ND REDUCES REDUNDANCY?

Let $\mathcal{D}^i(v)$ denote the set of nodes at exactly $i$ hops from $v$, and let $\Delta$ be the average node degree. Then we approximate

**Algorithm 1** CMV-ND

---

1: **Input:** Graph $\mathcal{G}$, target node $v$ and hop distance $k$
2: **Output:** Set of $k$-differential hop neighborhood $\mathcal{D}^k(v)$
3: Initialize visited set: $\mathcal{V}_{\text{visited}} \leftarrow \emptyset$
4: Initialize priority queue: $PQ \leftarrow []$
5: Push $(0, v)$ into $PQ$
6: Initialize differential hop set: $\mathcal{D}^k(v) \leftarrow \emptyset$
7: **while** priority queue $PQ$ is not empty **do**
8:     Pop $(hops, u)$ from $PQ$
9:     **if** $hops > k$ **then**
10:         **break**
11:     **end if**
12:     **if** $hops = k$ and $u \neq v$ **then**
13:         Add node $u$ to $\mathcal{D}^k(v)$
14:     **end if**
15:     **if** $hops < k$ **then**
16:         **for** each neighbor $w$ of $u$ **do**
17:             **if** $w$ not in $\mathcal{V}_{\text{visited}}$ **then**
18:                 Add $w$ to $\mathcal{V}_{\text{visited}}$
19:                 Push $(hops + 1, w)$ into $PQ$
20:             **end if**
21:         **end for**
22:     **end if**
23: **end while**
24: **Return** the set of differential hop neighbors $\mathcal{D}^k(v)$

---

$|\mathcal{D}^i(v)| \approx \Delta^i$. In a $k$-layer GNN, each $\mathcal{D}^i(v)$ contributes to all layers up to $k$, leading to repeated accesses. The total number of feature accesses is:

$$\sum_{i=1}^{k}(k - i + 1)|\mathcal{D}^i(v)| \approx \sum_{i=1}^{k}(k - i + 1)\Delta^i, \quad (6)$$

where $\Delta^i$ denotes the number of $i$-hop neighbors accessed per node. In contrast, CMV-ND accesses each differential hop neighborhood exactly once, resulting in:

$$\sum_{i=1}^{k}|\mathcal{D}^i(v)| \approx \sum_{i=1}^{k}\Delta^i. \quad (7)$$

We define the redundancy ratio $R_{\text{red}}$ as the ratio between the total number of feature accesses in a traditional GNN and the number of neighbor nodes actually used in CMV-ND:

$$R_{\text{red}} = \frac{\sum_{i=1}^{k}(k - i + 1)\Delta^i}{\sum_{i=1}^{k}\Delta^i}, \quad (8)$$

The ratio is always greater than 1 for $k \geq 2$ and $\Delta > 1$, indicating that traditional GNNs repeatedly access neighbor features across layers, while CMV-ND avoids such redundancy. Moreover, the redundancy ratio increases with larger $k$ and $\Delta$, illustrating that the inefficiency of traditional message passing becomes more pronounced in deeper networks or denser graphs.

## 3.4. Complexity Analysis

The overall time complexity of *CMV-ND* is $\mathcal{O}(n\Delta^k)$, where $n$ is the number of nodes. This arises from performing the RNS for each node, expanding its neighborhood up to $k$ hops. The NDS has a time complexity of $\mathcal{O}(k)$ per node. Regarding memory, *CMV-ND* requires $\mathcal{O}(n)$ space to store the visited nodes and the priority queue, leading to an overall memory complexity of $\mathcal{O}(n)$. Given that large-scale graphs are typically sparse, with $\Delta \ll n$, and *CMV-ND* only requires one-time preprocessing, it is highly scalable for large graphs.

### 3.4.1. TIME COMPLEXITY

**Recursive Neighborhood Search.** The algorithm starts from each node $v$ and recursively expands its neighborhood by incorporating nodes that are exactly $k$ hops away at each step. At each hop, the number of newly encountered nodes is determined by the degrees of the nodes reached at the previous hop. Specifically, the number of nodes added at each hop is bounded by the average degree $\Delta^i$, where $\Delta$ is the maximum degree of any node in the graph. Thus, the total number of nodes explored for each node $v$ up to the $k$-th hop is $\sum_{i=0}^{k}\Delta^i = 1 + \Delta + \Delta^2 + \cdots + \Delta^k = \mathcal{O}(\Delta^k)$. Since the search is performed for each of the $n$ nodes in the graph, the overall time complexity of the RNS for the entire graph is $\mathcal{O}(n\Delta^k)$.

**Neighborhood Differential Strategy.** The $k$-differential hop neighborhood for each node is computed as $\mathcal{D}^k(v) = \mathcal{N}^k(v) \setminus \mathcal{N}^{k-1}(v)$. Since set differences can be calculated in constant time for each pair of neighborhoods, the time complexity for this step is $\mathcal{O}(nk)$.

### 3.4.2. MEMORY COMPLEXITY

The memory complexity of *CMV-ND* is $\mathcal{O}(n)$, which arises from maintaining two primary data structures: 1) a set of visited nodes, which tracks the nodes that have already been explored to avoid redundant computations and consumes $\mathcal{O}(n)$ memory; 2) a priority queue used to manage the frontier nodes during neighborhood expansion. Since each node is added at most once, the memory usage of the priority queue is also $\mathcal{O}(n)$.

## 4. Experiment

In this section, we provide a comprehensive evaluation of our proposed *CMV-ND* by addressing the following questions. We do not conduct an ablation study, as *CMV-ND* is a plug-and-play paradigm whose components are inherently indivisible.

- **Q1: Superiority.** Does the *CMV-ND* paradigm outperform existing state-of-the-art DGC under attribute-

*Table 1.* Clustering performance comparison among various DGC methods on attribute-missing graphs. The reported results are the average clustering outcomes over ten runs on six graph datasets with 0.6 missing rate. "OOM" means out-of-memory on a 24 GB RTX 3090 GPU. "'w/o" and w/" denote the method without and with CMV-ND preprocessing, respectively.

| Dataset | CMV-ND | Metric | K-means | DGI | MVGRL | ProGCL | AGC-DRR | CCGC | S³GC | AMGC | DinkNet |
|---|---|---|---|---|---|---|---|---|---|---|---|
| Cora | w/o | ACC | 31.55±2.23 | 55.74±2.13 | 61.26±2.31 | 46.54±1.20 | 34.92±1.23 | 36.35±2.89 | 64.53±2.54 | 67.26±1.65 | 66.54±1.65 |
| | | NMI | 14.98±1.32 | 41.37±1.34 | 48.43±2.57 | 35.10±1.13 | 15.63±1.67 | 18.44±0.68 | 48.30±1.47 | 50.23±1.32 | 48.23±1.57 |
| | | ARI | 9.12±1.15 | 44.25±0.72 | 45.74±1.94 | 22.86±1.08 | 11.51±0.58 | 15.34±1.63 | 49.32±1.31 | 44.32±1.43 | 41.54±2.21 |
| | | F1 | 32.25±1.67 | 54.20±1.10 | 59.46±2.09 | 44.27±1.09 | 23.48±1.97 | 34.61±2.13 | 61.37±1.54 | 61.99±2.45 | 60.57±1.98 |
| | w/ | ACC | 45.55±2.05 | 61.74±1.93 | 66.36±1.31 | 51.14±1.24 | 48.41±1.29 | 43.48±1.98 | 70.34±1.53 | 69.26±1.45 | 72.54±1.29 |
| | | NMI | 21.94±1.12 | 48.37±1.65 | 67.43±2.57 | 43.10±1.51 | 31.73±1.74 | 27.44±1.34 | 53.14±1.12 | 54.23±1.32 | 54.29±0.77 |
| | | ARI | 25.12±1.15 | 51.25±1.54 | 56.34±2.97 | 44.86±1.73 | 21.51±0.78 | 29.34±1.51 | 51.07±2.04 | 48.32±1.43 | 51.54±0.57 |
| | | F1 | 47.19±2.23 | 60.57±1.49 | 67.78±1.94 | 51.61±1.51 | 54.48±1.47 | 44.72±1.83 | 69.77±1.53 | 62.32±1.45 | 70.57±0.64 |
| CiteSeer | w/o | ACC | 38.27±1.33 | 53.60±1.31 | 57.20±0.58 | 58.85±0.93 | 53.40±1.05 | 44.80±2.13 | 57.61±1.51 | 61.23±1.87 | 58.34±1.27 |
| | | NMI | 17.95±2.28 | 35.54±1.24 | 34.12±2.40 | 34.96±1.21 | 31.47±2.85 | 29.81±0.56 | 34.59±1.40 | 34.02±1.27 | 32.87±1.41 |
| | | ARI | 14.32±2.91 | 36.78±1.67 | 27.62±0.78 | 30.11±1.26 | 34.48±2.26 | 28.07±0.57 | 35.14±2.12 | 35.47±2.48 | 33.96±1.22 |
| | | F1 | 31.25±1.34 | 49.98±0.97 | 53.85±2.33 | 42.97±2.07 | 57.21±0.65 | 36.96±0.54 | 55.91±1.31 | 57.34±2.34 | 53.41±1.15 |
| | w/ | ACC | 49.27±1.32 | 60.60±1.47 | 57.20±0.58 | 62.15±0.79 | 57.41±1.51 | 53.98±1.93 | 67.17±1.47 | 65.71±2.83 | 68.56±1.71 |
| | | NMI | 29.15±2.28 | 36.54±0.84 | 36.12±1.40 | 39.64±1.54 | 36.41±2.85 | 38.81±0.56 | 39.98±1.14 | 38.14±1.42 | 40.87±1.11 |
| | | ARI | 27.32±1.94 | 43.65±1.41 | 29.98±1.21 | 36.77±1.47 | 37.48±1.26 | 34.45±0.51 | 40.54±1.34 | 39.37±1.68 | 41.61±1.72 |
| | | F1 | 42.31±2.04 | 57.67±1.47 | 57.84±1.34 | 55.97±1.91 | 61.21±0.61 | 44.96±0.31 | 60.94±2.44 | 59.64±2.34 | 61.31±1.54 |
| Amazon-Photo | w/o | ACC | 28.29±1.94 | 38.06±1.74 | 37.85±1.19 | 35.55±2.19 | 70.48±2.10 | 54.23±0.61 | 69.99±0.43 | 72.37±0.94 | 70.75±1.47 |
| | | NMI | 14.24±1.31 | 28.64±1.44 | 22.87±1.50 | 28.43±2.37 | 59.63±1.40 | 39.42±1.47 | 63.34±1.41 | 66.53±1.81 | 64.36±1.28 |
| | | ARI | 4.52±1.03 | 18.46±2.13 | 10.21±1.26 | 25.92±1.20 | 52.42±1.33 | 36.47±1.57 | 59.93±1.61 | 60.15±1.43 | 58.13±1.34 |
| | | F1 | 22.96±2.06 | 29.78±1.71 | 27.14±0.93 | 26.51±0.62 | 66.87±1.46 | 51.27±1.28 | 62.34±1.77 | 68.03±1.64 | 63.91±1.24 |
| | w/ | ACC | 36.29±1.25 | 43.26±2.74 | 45.45±1.39 | 40.51±1.19 | 73.18±1.04 | 61.23±0.98 | 76.41±0.53 | 75.37±1.67 | 77.75±1.34 |
| | | NMI | 27.13±1.31 | 43.64±1.34 | 29.87±1.42 | 43.43±1.37 | 64.43±1.07 | 51.32±1.74 | 69.43±1.47 | 70.53±2.81 | 71.54±1.63 |
| | | ARI | 19.12±1.97 | 25.46±1.83 | 34.21±2.26 | 41.27±1.27 | 56.42±1.23 | 44.71±1.45 | 65.43±1.37 | 64.37±2.43 | 66.51±2.71 |
| | | F1 | 34.96±1.22 | 36.54±1.55 | 43.24±1.93 | 36.41±0.57 | 68.47±1.39 | 57.65±1.47 | 71.34±1.39 | 70.03±0.84 | 72.03±1.43 |
| Reddit | w/o | ACC | 8.45±2.15 | 18.88±1.79 | OOM | 62.23±1.62 | OOM | OOM | 66.78±1.94 | OOM | 66.54±2.14 |
| | | NMI | 12.30±1.76 | 26.41±1.04 | OOM | 64.17±1.13 | OOM | OOM | 67.39±1.74 | OOM | 68.95±1.70 |
| | | ARI | 2.90±1.98 | 15.34±0.74 | OOM | 60.24±1.47 | OOM | OOM | 62.34±1.22 | OOM | 61.34±1.46 |
| | | F1 | 6.80±2.61 | 17.48±0.57 | OOM | 47.73±1.84 | OOM | OOM | 56.43±1.34 | OOM | 57.98±2.22 |
| | w/ | ACC | 28.45±1.15 | 23.98±1.41 | OOM | 65.23±1.41 | OOM | OOM | 68.98±0.74 | OOM | 69.31±1.84 |
| | | NMI | 10.40±1.36 | 33.45±1.47 | OOM | 67.17±1.31 | OOM | OOM | 71.64±1.37 | OOM | 71.95±1.01 |
| | | ARI | 14.91±1.45 | 24.35±0.84 | OOM | 63.24±1.97 | OOM | OOM | 64.37±1.17 | OOM | 65.41±1.62 |
| | | F1 | 26.80±2.41 | 23.55±2.43 | OOM | 53.77±1.40 | OOM | OOM | 60.97±1.34 | OOM | 62.87±1.32 |
| ogbn-arXiv | w/o | ACC | 18.11±2.31 | 23.88±1.79 | OOM | 24.94±1.16 | OOM | OOM | 31.98±1.97 | OOM | 33.74±1.47 |
| | | NMI | 21.32±2.54 | 35.41±1.34 | OOM | 30.37±0.98 | OOM | OOM | 31.07±1.27 | OOM | 32.74±1.24 |
| | | ARI | 7.67±1.45 | 11.34±1.94 | OOM | 23.03±2.61 | OOM | OOM | 24.71±1.31 | OOM | 25.43±0.97 |
| | | F1 | 13.94±1.84 | 23.48±1.91 | OOM | 14.61±2.98 | OOM | OOM | 21.33±1.14 | OOM | 22.99±0.64 |
| | w/ | ACC | 34.11±1.34 | 27.81±1.79 | OOM | 29.94±1.27 | OOM | OOM | 35.77±1.07 | OOM | 36.84±2.04 |
| | | NMI | 27.32±1.94 | 36.21±1.57 | OOM | 32.77±0.78 | OOM | OOM | 40.88±1.39 | OOM | 41.74±1.07 |
| | | ARI | 20.67±2.43 | 14.54±2.14 | OOM | 27.07±1.41 | OOM | OOM | 28.71±1.47 | OOM | 30.43±1.84 |
| | | F1 | 19.94±1.84 | 21.86±1.91 | OOM | 19.81±1.98 | OOM | OOM | 24.83±1.31 | OOM | 25.91±1.67 |
| ogbn-products | w/o | ACC | 19.23±1.64 | 29.78±1.89 | OOM | 26.30±2.43 | OOM | OOM | 30.98±1.74 | OOM | 31.09±0.94 |
| | | NMI | 22.41±0.86 | 40.41±2.34 | OOM | 41.37±1.03 | OOM | OOM | 41.51±2.24 | OOM | 42.71±0.75 |
| | | ARI | 5.11±1.32 | 10.34±2.07 | OOM | 12.41±1.36 | OOM | OOM | 18.27±1.33 | OOM | 18.05±1.11 |
| | | F1 | 6.52±2.43 | 14.61±1.04 | OOM | 13.43±2.65 | OOM | OOM | 17.51±1.34 | OOM | 15.34±1.43 |
| | w/ | ACC | 25.21±1.42 | 33.54±1.64 | OOM | 31.30±2.43 | OOM | OOM | 34.57±0.43 | OOM | 35.91±0.43 |
| | | NMI | 21.91±0.87 | 42.41±2.74 | OOM | 41.37±2.03 | OOM | OOM | 44.31±2.37 | OOM | 45.71±0.79 |
| | | ARI | 15.41±1.59 | 19.34±1.87 | OOM | 13.41±2.36 | OOM | OOM | 19.89±1.46 | OOM | 20.05±0.44 |
| | | F1 | 14.52±1.43 | 22.87±1.17 | OOM | 18.53±1.37 | OOM | OOM | 22.93±1.64 | OOM | 23.41±1.51 |

missing graph?

- **Q2: Sensitivity.** How sensitive is *CMV-ND* to the hyperparameter $K$, the number of views?

- **Q3: Efficiency.** What are the time and memory costs of *CMV-ND* on large-scale graphs?

- **Q4: Robustness.** Although *CMV-ND* is designed under the assumption of attribute missingness, how does it perform in clustering when the attributes are complete?

- **Q5: Effectiveness.** How does the performance of using hop representations derived from propagation methods as multi-view?

**Q1–Q3** are discussed in Section 4.2–4.4. Due to space limitations, we provide the answers to **Q4** and **Q5** in Appendix D.2 and Appendix D.3.

### 4.1. Experimental Setup

**Datasets.** To evaluate clustering performance, we use six attribute-based graph datasets, including three small graphs: Cora, CiteSeer, and Amazon-Photo, and three large graphs: Reddit, ogbn-arXiv, and ogbn-products.

**Implementation Details.** To ensure a fair comparison, we conduct 10 experimental iterations under identical conditions and report the average results. All experiments are performed on a system equipped with a 24GB RTX 3090

*Table 2.* The performance comparison of six MVC methods on attribute-missing graphs is presented. The reported results are the average clustering outcomes over ten runs on six graph datasets. We use the $K + 1$ views obtained by *CMV-ND* as input. The experiment is conducted on graphs with a 0.6 missing rate, after leveraging FP imputation. "OOM" means the out-of-memory failure on 24GB RTX 3090 GPU.

| Dataset | Metric | MFLVC | MCMVC | DIMVC | SDMVC |
|---------|--------|-------|-------|-------|-------|
| Cora | ACC | 48.37±1.75 | 54.74±1.13 | 61.26±2.31 | 48.23±1.54 |
| | NMI | 33.23±1.67 | 39.37±1.33 | 55.43±2.57 | 39.06±1.67 |
| | ARI | 35.41±1.21 | 41.14±1.54 | 50.74±1.94 | 25.78±1.75 |
| | F1 | 45.29±2.19 | 53.20±2.31 | 65.46±2.09 | 48.51±1.89 |
| CiteSeer | ACC | 51.37±1.33 | 51.60±1.74 | 57.20±0.58 | 61.45±1.34 |
| | NMI | 31.25±1.38 | 33.44±1.77 | 34.12±2.40 | 36.56±1.06 |
| | ARI | 29.55±2.34 | 34.51±1.84 | 27.62±0.78 | 33.54±1.89 |
| | F1 | 47.57±1.67 | 45.48±1.37 | 53.85±2.33 | 44.84±1.87 |
| Amazon -Photo | ACC | 42.49±2.25 | 38.06±1.74 | 69.85±2.19 | 38.65±1.35 |
| | NMI | 30.23±1.51 | 24.44±1.31 | 64.87±1.50 | 31.51±1.46 |
| | ARI | 24.12±2.17 | 19.31±1.64 | 58.21±1.26 | 27.41±1.45 |
| | F1 | 32.96±1.61 | 27.89±1.39 | 60.14±0.93 | 28.45±1.45 |
| Reddit | ACC | 31.55±2.11 | 15.54±1.69 | 64.23±2.13 | 62.31±1.93 |
| | NMI | 17.20±1.76 | 24.31±1.32 | 66.35±1.71 | 63.25±1.04 |
| | ARI | 29.91±2.45 | 13.41±0.54 | 59.23±1.46 | 58.26±1.26 |
| | F1 | 18.80±1.41 | 19.46±0.67 | 54.28±1.42 | 51.33±1.65 |
| ogbn -arXiv | ACC | 47.21±1.78 | 25.87±1.89 | 28.44±1.44 | 27.84±0.56 |
| | NMI | 29.12±1.64 | 24.53±1.47 | 27.78±1.56 | 25.37±0.98 |
| | ARI | 19.47±1.33 | 14.54±1.76 | 28.67±0.73 | 24.13±1.31 |
| | F1 | 23.74±1.64 | 21.78±1.43 | 23.15±0.89 | 18.98±2.98 |
| ogbn -products | ACC | 23.21±1.34 | 26.86±1.54 | 28.09±0.41 | 24.30±1.33 |
| | NMI | 27.41±0.56 | 37.41±1.44 | 39.71±0.32 | 38.37±1.31 |
| | ARI | 13.51±1.79 | 19.21±1.84 | 16.15±1.67 | 15.31±1.41 |
| | F1 | 18.42±1.75 | 25.96±1.79 | 13.58±1.75 | 14.79±1.46 |

GPU and 64GB RAM. We evaluate clustering performance using four widely adopted metrics (Liu et al., 2023b), including Accuracy (ACC), Normalized Mutual Information (NMI), Adjusted Rand Index (ARI), and F1-score (F1). Notably, ARI ranges from -1 to 1, while the other metrics range from 0 to 1. All experiments are implemented using Python 3.9 and PyTorch 1.12. Unless otherwise specified, we set the number of propagation hops to $K = 7$ and the missing attribute rate to 0.6. For fair comparison, all downstream clustering methods follow the default hyperparameter configurations used in their original implementations. The number of clusters is set to the ground-truth number of classes for each dataset. Additional dataset statistics and descriptions are provided in Table 4.

**Baselines.** To demonstrate the superiority of the *CMV-ND* paradigm, we construct a set of clustering methods based on *CMV-ND*, which includes a series of DGC and MVC methods. The implementation details can be found in Appendix B. Specifically, for classical clustering, we employ K-Means, which utilizes the concept of anomaly maximization to separate samples. DGC methods leverage GNNs to uncover graph structures, followed by grouping nodes into distinct clusters. These methods include DGI (Velickovic et al., 2019), MVGRL (Hassani & Khasahmadi, 2020), ProGCL (Xia et al., 2022), AGC-DRR (Gong et al., 2022), CCGC (Yang et al., 2023a), S³GC (Devvrit et al.,

2022), AMGC (Tu et al., 2024a), and Dink-Net (Liu et al., 2023b). The MVC methods exploit both consistency and complementarity in view group nodes, including MFLVC (Xu et al., 2022c), MCMVC (Geng et al., 2024), DIMVC (Xu et al., 2022a), and SDMVC (Xu et al., 2022b).

**4.2. Clustering Performance Comparation (Q1)**

Since our method does not involve feature imputation for attribute-missing nodes, we preprocess all methods with FP (Park et al., 2022) for feature completion to evaluate clustering performance on attribute-missing graphs. The clustering performance of methods based on the *CMV-ND* paradigm and the comparison methods is summarized in Tables 1 and Table 2. The missing rate design follows the setting of the method AMGC (Tu et al., 2024a), which uses a missing rate of 0.6. From the results, we can draw the following conclusions.

- The previous state-of-the-art method for attribute-missing graph clustering, AMGC, suffers from out-of-memory (OOM) failures when applied to large graphs, limiting its scalability. Specifically, AMGC fails to complete training on Reddit, ogbn-arXiv, and ogbn-products, while DinkNet combined with *CMV-ND* completes all runs on these datasets with stable memory usage.

- When our *CMV-ND* is combined with Dink-Net, it not only achieves superior performance on large graphs with missing attributes but also outperforms existing SOTA methods on smaller graphs. For example, on Cora, it yields ACC 72.54% and F1 70.57%, surpassing all baselines; on Reddit, it achieves the best ACC 69.31% and F1 62.87% among all methods that do not encounter OOM.

- Comparing the performance of *CMV-ND* before and after leveraging it to DGC shows that our approach does not result in any negative impact under any circumstance. Across all datasets and metrics, the incorporation of *CMV-ND* either improves or maintains performance. Notably, the average ACC improves by 13.8% on Cora and 5.8% on Amazon-Photo.

- While the MVC implementation of *CMV-ND* is weaker than its DGC counterpart, all MVC methods can be applied to large graphs. For instance, DIMVC reaches 69.85% ACC and 60.14% F1 on Amazon-Photo, and 64.23% ACC on Reddit, while maintaining feasibility on ogbn-arXiv and ogbn-products. In the future, we plan to design an MVC method specifically tailored for *CMV-ND* to further enhance its performance in MVC.

In addition, to intuitively illustrate the clustering quality improvements brought by *CMV-ND*, we employ 2D t-SNE

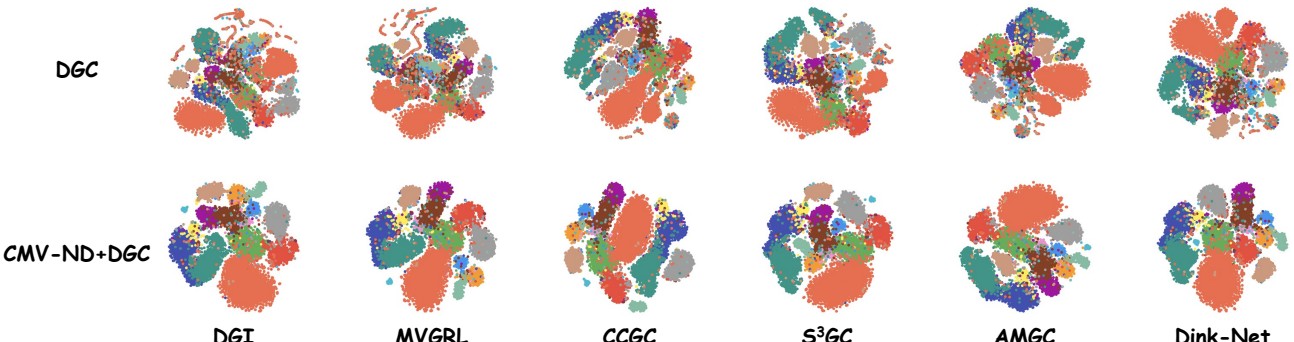

*Figure 2.* T-SNE visualization of node representations generated by six methods on the Co.CS dataset.

([Van der Maaten & Hinton](), 2008) to visualize the node representations generated by six baseline algorithms before and after leveraging our *CMV-ND*. As shown in Figure 2, *CMV-ND* substantially improves the latent space structure: the orange cluster becomes more compact and coherent, while the purple cluster exhibits fewer cross-class intrusions.

### 4.3. Hyper-Parameters Analysis (Q2)

To further evaluate the performance of *CMV-ND*, we specifically investigate the effect of the hyperparameter $K$ on model results. The experimental setup follows Section 4.2, where clustering performance is evaluated using the state-of-the-art combination of *CMV-ND* and Dink-Net.

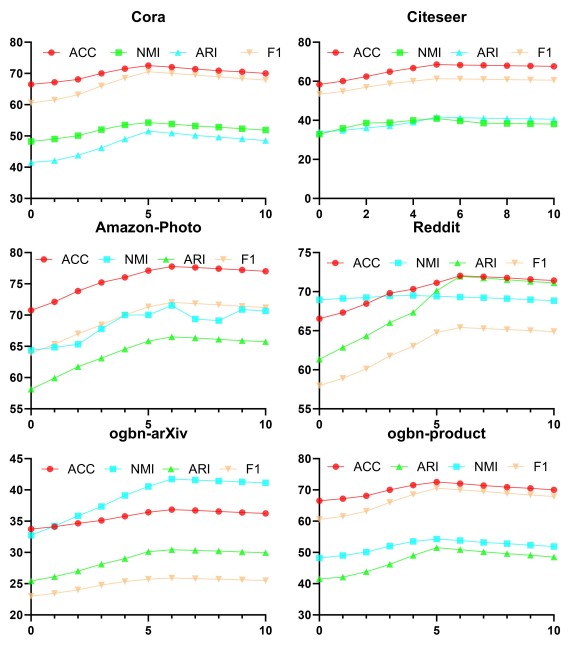

*Figure 3.* The hyperparameter analysis of *CMV-ND*, where the horizontal axis denotes the number of differential hops $K$. Notably, $K = 0$ corresponds to the original Dink-Net without *CMV-ND*.

As shown in Figure 3, We observe that for all datasets, the method is insensitive to the value of $K$. In all cases, *CMV-ND* consistently provides a positive gain to Dink-Net. Moreover, the highest performance is achieved when $K$ is around $(5, 7)$. Larger values of $K$ do not significantly degrade performance, indicating that our *CMV-ND* does not suffer from the over-smoothing issue commonly observed in traditional graph propagation paradigms.

### 4.4. Time and Memory Consumption (Q3)

This section is used to answer the time and memory costs of *CMV-ND*. Since *CMV-ND* is a preprocessing step, we do not compare it with other methods but instead report its time and memory usage. As shown in Table 3, the preprocessing time and memory consumption of *CMV-ND* vary across datasets. The costs remain acceptable across all datasets. Since *CMV-ND* is a preprocessing step, it only needs to be executed once, making its computational overhead negligible in the long run. Moreover, *CMV-ND* can be performed on CPUs and memory without relying on GPU resources, ensuring its scalability to large-scale graphs.

*Table 3.* Preprocessing time and memory consumption of *CMV-ND* on different datasets. The table reports the time cost (in seconds) and CPU memory consumption (in MB) for the preprocessing step of *CMV-ND*.

| Dataset | Time Cost (s) | CPU Memory Cost (MB) |
|---|---|---|
| **Cora** | 5.585 | 84.59 |
| **CiteSeer** | 3.311 | 183.3 |
| **Amazon-Photo** | 13.445 | 247.67 |
| **Reddit** | 189.34 | 446.64 |
| **ogbn-arXiv** | 144.23 | 270.46 |
| **ogbn-products** | 274.57 | 672.84 |

## 5. Conclusion and Future Work

In this work, we propose a novel paradigm, *CMV-ND*, to address the challenges of large-scale graph clustering under attribute-missing conditions. Our analysis reveals that

the key to tackling this challenge lies in effectively leveraging the graph structure. Unlike the typical "aggregate-encode-predict" pipeline of GNNs, *CMV-ND* does not rely on propagation operations to utilize the graph structure. Instead, it directly encodes graph structure information into the views through differential neighborhoods, enabling efficient handling of large-scale graph data without the need for sampling. Experimental results on six widely-used graph datasets demonstrate that *CMV-ND* significantly improves the clustering performance of various DGC methods on attribute-missing graphs. Moreover, *CMV-ND* naturally bridges the gap between DGC and MVC. While the method achieves promising results, its current implementation in the MVC setting shows slightly weaker performance. As part of future work, we aim to develop MVC methods tailored for *CMV-ND*, with the goal of further enhancing its adaptability, model complexity, and performance in real-world applications.

## Acknowledgments

This work is supported by the National Natural Science Foundation of China (No. 62441618, No. 62325604, and No. 62276271), the Natural Science Foundation of Hainan University (No. XJ2400009401).

## Impact Statement

This paper presents work whose goal is to advance the field of Machine Learning.

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

# Appendix of "Scalable Attribute-Missing Graph Clustering via Neighborhood Differentiation"

## A. Notations & Datasets

The basic notations are outlined in Table 4. These notations provide a formal foundation for our study, ensuring clarity and consistency in the subsequent discussions.

*Table 4.* The notations.

| Notation | Meaning |
|---|---|
| $\mathbf{A}$ | Adjacency Matrix |
| $\mathcal{V}$ | Set of Vertex |
| $\mathcal{E}$ | Set of Edges |
| $d$ | Feature Dimension |
| $\widetilde{\mathcal{G}}$ | Attribute-missing Graph |
| $\Delta$ | Maximum degree of any node in the graph |
| $\mathcal{D}^k(v)$ | $k$-differential hop neighborhood |
| $\mathcal{N}^k(v)$ | $k$-hop neighborhood |
| $S_a^{(k)}$ | State at the $k$-th hop |
| $\mathbf{H} \in \mathbb{R}^{N \times d}$ | Node Attribute Matrix |
| $\mathbf{h}^k(v) \in \mathbb{R}^d$ | $k$-differential hop neighborhood representation |

Table 5 summarizes the key statistics of the six datasets used in this study. These datasets span a wide range of sizes and characteristics. For instance, CiteSeer includes 3,327 nodes and 4,614 edges, while Amazon-Photo consists of 7,650 nodes and 119,081 edges. Furthermore, the graph densities also differ significantly. As an example, Cora has a density of 0.07%, whereas Amazon-Photo exhibits a density of 0.25%.

*Table 5.* Statistics of six datasets.

| Dataset | Type | # Nodes | # Edges | # Feature Dims | # Classes |
|---|---|---|---|---|---|
| Cora | Citation Graph | 2,708 | 5,278 | 1,433 | 7 |
| CiteSeer | Citation Graph | 3,327 | 4,614 | 3,703 | 6 |
| Amazon-Photo | Co-Purchase Graph | 7,650 | 119,081 | 745 | 8 |
| ogbn-arxiv | Citation Graph | 169,343 | 1,166,243 | 128 | 40 |
| Reddit | Social Network Graph | 232,965 | 23,213,838 | 602 | 41 |
| ogbn-products | Co-Purchase Graph | 2,449,029 | 61,859,140 | 100 | 47 |

## B. How to Leverage Multi-View Representations for Graph Clustering?

In this section, we demonstrate how multi-view representations can be applied to two graph clustering methods: DGC and MVC. We focus on proving the consistency and complementarity of multi-view representations in these tasks.

**DGC.** The multi-view representations can be directly fused to form a unified node representation for clustering. For example, we concatenate the feature vectors from each view, resulting in a combined representation. Let $\mathbf{h}_v^{(k)}$ be the feature vector for node $v$ in the $k$-hop neighborhood. The fused representation $\mathbf{h}_v^{\text{fuse}}$ can be written as:

$$\mathbf{h}_v^{\text{fuse}} = \text{concat}(\mathbf{h}_v^{(0)}, \mathbf{h}_v^{(1)}, \ldots, \mathbf{h}_v^{(k)}), \tag{9}$$

where $\mathbf{h}_v^{(i)} \in \mathbb{R}^d$ represents the feature vector of node $v$ in the $i$-th neighborhood view. This fused representation can then be directly used as input for DGC.

**MVC.** MVC algorithms typically rely on the assumption that different views provide complementary information while maintaining consistency. Our multi-view representations naturally satisfy these conditions.

- Consistency. The consistency of the multi-view representation is grounded in the homophily assumption, which posits that nodes with similar attributes tend to be connected in the graph. Given this assumption, nodes that are close in the graph, i.e., those with similar local structures, will have similar representations across different views. Formally, for nodes $v$ and $u$, if $\mathcal{D}^k(v) = \mathcal{D}^k(u)$ for some $k$, then the multi-view representations $\mathbf{h}_v^{(i)}$ and $\mathbf{h}_u^{(i)}$ across different views $i$ will be close:

$$\|\mathbf{h}_v^{(i)} - \mathbf{h}_u^{(i)}\|_2 \approx 0 \quad \text{for all} i \in \{0, 1, \ldots, k\}. \tag{10}$$

- Complementarity. The complementarity of the multi-view representations arises from two key aspects. On the one hand, different hops neighborhood, such as the $k$-hop neighborhood, capture node relationships from different perspectives. Specifically, the $k$-differential hop neighborhood captures local structural information, while the $(k + 1)$-differential hop neighborhood helps understand the global relationships between nodes. On the other hand, Eq. (2) ensures that each neighborhood view contributes non-overlapping information to the multi-view representation, thereby providing complementary insights into the position of the node within the graph.

# C. Additional Relate Work

## C.1. Deep Graph Clustering

This section discusses recent advances in deep graph clustering (DGC) for attribute-complete graphs, where existing methods can be broadly categorized into two main paradigms: (1) graph autoencoder-based methods (reconstruction models) and (2) contrastive Learning-based methods.

**Graph Autoencoder-based Methods.** To effectively utilize both graph structure and node attributes, many existing approaches employ autoencoder-based architectures (Tu et al., 2024b; 2025b). These methods typically use GNNs to encode node representations, aiming to reconstruct the graph structure via the inner product of the learned embeddings. Notable examples include graph auto-encoder (GAE) and variational graph auto-encoder (VGAE), which learn low-dimensional representations to capture latent graph structures, thereby improving clustering performance. Building upon the autoencoder framework, graph adversarial learning approaches (GALA) (Park et al., 2019), adversarially regularized graph autoencoder (ARGA), and adversarially regularized variational graph autoencoder (ARVGA) (Pan et al., 2019) introduce Laplacian sharpening and generative adversarial learning to enhance representation learning. Further extending this paradigm, deep fusion clustering network (DFCN) (Tu et al., 2021) integrates representations learned from both autoencoders and graph autoencoders for more robust consensus representation learning.

**Contrastive Learning-based Methods.** Contrastive learning has emerged as a powerful paradigm for graph clustering. deep graph infomax (DGI) (Velickovic et al., 2019) improves embedding representations by maximizing mutual information between global and local graph structures. InfoGraph (Sun et al., 2019) extends DGI by learning unsupervised representations at the graph level. Multi-view graph representation Learning (MVGRL) (Hassani & Khasahmadi, 2020) further enhances these ideas by leveraging node diffusion and contrasting node representations across augmented graph views. More recent advancements include graph contrastive learning and enhanced (GRACE) (Zhu et al., 2020), which maximizes node embedding consistency between corrupted graph views. Bootstrapped graph latents (BGRL) (Thakoor et al., 2022) builds upon the bootstrap your own latent (BYOL) (Grill et al., 2020) framework, introducing self-supervised learning to the graph domain while eliminating the need for negative sampling. Moreover, Liu et al. (2022a; 2024a) design the dual correlation reduction strategy in the DCRN model to alleviate the representation collapse problem. Besides, HSAN (Liu et al., 2023d) mines the hard sample pairs via the dynamic weighting strategy. And SCGC (Liu et al., 2023c) simplifies the graph augmentation with parameter-unshared Siamese encoders and embedding disturbance. Despite effectiveness, contrastive

learning-based methods are computationally expensive due to their reliance on graph augmentations and complex objective functions, making them difficult to scale to large datasets. A scalable method termed Dink-Net (Liu et al., 2023b) is proposed to solve this problem. Besides, (Liu et al., 2023a) is presented to solve the problem of the unknown number of clusters via reinforcement learning. Benefiting from these, deep graph clustering methods are applied to practical applications like recommendations (Liu et al., 2024b;c). More details can be found in this survey paper (Liu et al., 2022b).

### C.2. Large-Scale Graph Learning

In recent years, numerous scalable GNN methods have been developed to address the challenges of large-scale graph data processing. Graph sample and aggregation (GraphSAGE) (Hamilton et al., 2017) introduces an inductive learning framework that efficiently processes large graphs by sampling and aggregating node features from local neighborhoods. Fast graph convolutional network (FastGCN) (Chen et al., 2018) further improves efficiency by performing node sampling independently at each layer, effectively reducing computational and memory overhead. To enhance scalability, simplified graph convolution (SGC) (Wu et al., 2019) decouples transformation and propagation in graph convolutional network (GCN), significantly improving computational efficiency. GraphSaint (Zeng et al., 2019) and clustered graph convolutional network (Cluster-GCN) (Chiang et al., 2019) maintain graph structure integrity through subgraph sampling, enabling more effective large-scale graph processing.

### C.3. Attribute-Missing Graph Completion

The objective of attribute-missing graph completion is to enable effective learning on graphs where node attributes are partially or entirely missing. This problem has garnered significant attention, and existing approaches can be broadly categorized into two paradigms: (1) gcn-based models and (2) feature completion methods.

**GCN-based Models.** GCN-based models aim to directly encode graphs with missing attributes while simultaneously restoring the missing information during the learning process. These methods typically leverage GCNs to handle incomplete attribute data (Tu et al., 2022; 2025a). GCN for Missing Features (GCNMF) (Taguchi et al., 2021) introduces a Gaussian Mixture Model (GMM) to adjust GCNs for attribute-missing graphs. Similarly, Partial GNN (PaGNN) (Jiang & Zhang, 2020) employs a partial message passing (MP) scheme that integrates known features during propagation. While these methods perform well in semi-supervised settings, they rely on label supervision to implicitly restore missing attributes. As a result, they are not applicable to unsupervised tasks such as graph clustering, where label information is unavailable.

**Feature Completion Methods.** Feature completion methods restore missing node attributes as a preprocessing step, without relying on GCN encoding or parameterized learning. To eliminate label dependence, methods such as feature propagation (FP) (Park et al., 2022) and pseudo-confidence feature imputation (PCFI) (Um et al., 2023) leverage the graph topology to reconstruct missing attributes. FP propagates known features across the graph to impute missing values, while PCFI introduces channel confidence, assigning reliability scores to inferred features to improve robustness.

### C.4. Graph Structure Learning and Search Methods

Graph structure learning (GSL) and structure refinement have emerged as promising directions for enhancing graph learning tasks by optimizing or inferring the adjacency matrix. Recent works such as SUBLIME (Liu et al., 2022c), NodeFormer (Wu et al., 2022), and VIB-GSL (Sun et al., 2022) explore this avenue from different perspectives. SUBLIME adopts an unsupervised self-supervised contrastive learning framework to jointly learn graph structure and node embeddings for clustering tasks. NodeFormer designs a kernerlized gumbel-softmax operator for edge prediction, enabling structure refinement through pairwise node correlations. VIB-GSL introduces variational inference and information bottleneck principles to learn robust and generalizable graph structures.

Despite their effectiveness, these methods generally assume access to complete node attribute information to either guide similarity estimation or construct informative graph structures. In contrast, our work specifically targets the attribute-missing setting, where many nodes lack features entirely. Under such conditions, structure learning methods depending on pairwise feature similarity or global embedding consistency become unreliable or even infeasible.

# D. Additional Experimental Result

## D.1. Comparison with SAT, ITR, and SVGA

To further verify the generality and compatibility of *CMV-ND*, we conduct additional experiments comparing it with three representative methods tailored for attribute-missing graphs: SAT (Chen et al., 2020), ITR (Tu et al., 2022), and SVGA (Yoo et al., 2022). We evaluate their original performance as well as their variants enhanced with *CMV-ND* as a preprocessing step. As shown in Table 6, *CMV-ND* consistently improves clustering performance across all three base methods on Cora, Citeseer, and Amazon-Photo under 60% feature missing rate. In particular, notable gains are observed in both clustering accuracy and stability.

*Table 6.* Clustering performance (%) of SAT, ITR, and SVGA on attribute-missing graphs. All experiments use a 0.6 missing rate. Reported results are averages over 10 runs. OOM denotes out-of-memory failure on large-scale graphs.

| Dataset | Metric | SAT | SAT + CMV-ND | ITR | ITR + CMV-ND | SVGA | SVGA + CMV-ND |
|---|---|---|---|---|---|---|---|
| Cora | ACC | 54.31±1.07 | 63.42±1.11 | 39.87±1.12 | 52.13±1.04 | 45.56±1.94 | 55.41±1.29 |
| | NMI | 34.05±0.84 | 46.95±1.06 | 22.86±0.85 | 30.47±0.94 | 29.94±2.32 | 39.36±1.27 |
| | ARI | 27.14±0.92 | 41.68±1.14 | 8.41±0.63 | 18.87±1.08 | 18.72±1.62 | 29.34±1.41 |
| | F1 | 53.68±0.91 | 62.08±1.17 | 35.07±1.03 | 46.07±1.02 | 41.84±3.41 | 51.49±1.52 |
| Citeseer | ACC | 38.44±0.82 | 48.79±1.09 | 36.54±1.24 | 45.68±1.28 | 48.41±1.54 | 57.29±1.22 |
| | NMI | 14.01±0.35 | 25.85±0.98 | 19.58±1.11 | 29.01±1.32 | 26.91±1.38 | 33.14±1.47 |
| | ARI | 12.24±1.06 | 20.81±1.21 | 7.03±0.85 | 14.67±1.15 | 17.03±1.16 | 24.53±1.28 |
| | F1 | 36.89±0.76 | 47.41±1.06 | 34.64±1.29 | 46.32±1.19 | 45.41±2.02 | 56.88±1.35 |
| Amazon-Photo | ACC | 42.31±0.92 | 53.05±1.17 | 36.27±1.16 | 47.69±1.14 | 60.74±1.61 | 69.02±1.31 |
| | NMI | 48.13±0.65 | 56.23±0.93 | 40.18±1.22 | 49.58±1.25 | 64.89±0.82 | 72.19±1.14 |
| | ARI | 30.11±0.98 | 40.66±1.02 | 22.31±1.03 | 34.12±1.18 | 34.96±1.21 | 49.21±1.36 |
| | F1 | 36.04±0.94 | 45.32±1.09 | 31.75±1.02 | 42.41±1.23 | 52.91±2.98 | 63.82±1.29 |

## D.2. Clustering Performance on Attribute-Complete Graphs (Q4)

To evaluate the robustness of our paradigm, we also conduct experiments on attribute-complete graphs. As shown in Table 7, the results indicate that when node attributes are fully available, DGC methods incorporating *CMV-ND* exhibit nearly identical performance to their original counterparts. This suggests that our approach is non-intrusive and does not negatively impact clustering in attribute-complete settings. An exciting finding is that the classic K-means algorithm exhibits a remarkable performance improvement, even surpassing many DGC methods. We hypothesize that this is because conventional K-means does not leverage graph structures, whereas our *CMV-ND* effectively compensates for this limitation.

## D.3. Evaluating Multi-View Representations Derived from Propagation-Based Hops (Q5)

The proposed *CMV-ND* constructs multiple views for graph clustering by leveraging differential hop representations. Naturally, one might consider whether graph propagation paradigms can also generate multi-view representations in a similar manner. Specifically, we construct views using the multiplication of the adjacency matrix and the feature matrix:

Formally, given a graph $\mathcal{G} = (\mathcal{V}, \mathcal{E})$ with adjacency matrix $\mathbf{A}$ and node attribute matrix $\mathbf{H}$, we derive a series of views by leveraging the propagation operation iteratively:

$$\mathbf{H}^{(k)} = \mathbf{A}\mathbf{H}^{(k-1)}, \quad \text{with} \quad \mathbf{H}^{(0)} = \mathbf{H}, \tag{11}$$

where $\mathbf{H}^{(k)}$ represents the propagated node features after $k$ iterations. In this setup, each $\mathbf{H}^{(k)}$ can be regarded as a separate view, akin to the multi-view representations generated by *CMV-ND*.

Subsequently, we utilize these views as inputs to various DGC methods and evaluate their clustering performance. The experimental setup follows that of Section 4.2 to ensure consistency. As shown in Table 8, compared to the baseline results in Table 1, the propagation-based multi-view representations generally outperform the original DGC methods but remain inferior to *CMV-ND*. This observation further substantiates the effectiveness of modeling node hop representations as multi-views.

*Table 7.* Clustering performance comparison among various DGC methods on attribute-missing graphs. The reported results are the average clustering outcomes over ten runs on six graph datasets. "OOM" means out-of-memory on a 24 GB RTX 3090 GPU. "'w/o" and w/" denote the method without and with CMV-ND preprocessing, respectively.

| Dataset | CMV-ND | Metric | K-means | DGI | MVGRL | ProGCL | AGC-DRR | S³GC | DinkNet |
|---|---|---|---|---|---|---|---|---|---|
| Cora | w/o | ACC | 33.78±1.38 | 72.61±1.03 | 75.43±1.44 | 56.96±1.25 | 40.67±1.60 | 74.29±0.83 | 76.17±1.90 |
| | | NMI | 14.89±0.70 | 57.16±1.64 | 60.75±1.49 | 40.84±0.90 | 18.63±1.71 | 58.78±1.60 | 60.25±1.63 |
| | | ARI | 8.58±1.80 | 51.15±0.98 | 56.77±0.71 | 30.58±1.75 | 14.76±0.98 | 54.37±1.46 | 59.43±1.73 |
| | | F1 | 30.29±0.86 | 69.17±1.80 | 71.66±1.05 | 45.74±1.47 | 31.15±1.95 | 72.24±1.94 | 70.50±1.87 |
| | w/ | ACC | 74.00±1.46 | 71.69±1.24 | 75.31±1.59 | 57.07±1.38 | 41.25±1.89 | 73.98±0.84 | 75.86±1.93 |
| | | NMI | 57.10±0.73 | 57.76±1.59 | 61.42±1.24 | 40.64±1.19 | 18.93±1.97 | 58.65±1.67 | 59.96±1.48 |
| | | ARI | 52.31±0.41 | 51.33±0.85 | 56.20±0.82 | 29.98±2.01 | 14.12±1.27 | 53.98±1.35 | 59.68±1.58 |
| | | F1 | 72.50±1.33 | 69.22±2.04 | 72.48±1.08 | 44.86±1.49 | 31.45±1.70 | 72.21±2.15 | 70.95±1.94 |
| CiteSeer | w/o | ACC | 39.33±0.76 | 68.74±0.98 | 62.67±1.22 | 66.10±1.99 | 68.38±0.79 | 68.74±1.78 | 68.30±1.17 |
| | | NMI | 16.76±1.81 | 43.46±1.44 | 40.62±1.40 | 39.43±0.88 | 43.14±1.63 | 44.27±0.95 | 43.95±1.39 |
| | | ARI | 13.35±1.33 | 44.41±0.92 | 34.31±1.00 | 36.22±1.74 | 45.40±1.30 | 44.94±1.41 | 45.99±1.26 |
| | | F1 | 36.25±1.17 | 64.45±0.96 | 59.63±1.75 | 57.75±1.75 | 64.87±0.75 | 64.40±1.35 | 65.12±1.42 |
| | w/ | ACC | 68.11±1.27 | 68.04±1.08 | 63.00±1.07 | 65.90±1.88 | 68.18±0.73 | 67.92±1.87 | 67.82±1.08 |
| | | NMI | 43.45±0.67 | 43.07±1.30 | 41.17±1.22 | 40.12±0.80 | 42.77±1.53 | 43.58±1.12 | 44.12±1.52 |
| | | ARI | 43.47±0.35 | 44.63±0.94 | 34.82±1.09 | 36.29±1.57 | 45.40±1.17 | 45.30±1.34 | 46.35±1.29 |
| | | F1 | 59.06±0.66 | 63.83±0.77 | 59.39±1.92 | 58.62±1.81 | 64.91±0.82 | 63.90±1.51 | 64.98±1.56 |
| Amazon-Photo | w/o | ACC | 27.25±1.06 | 43.11±1.97 | 40.99±1.94 | 51.58±0.87 | 76.78±1.23 | 75.16±1.55 | 79.72±1.84 |
| | | NMI | 13.43±1.96 | 33.51±1.62 | 30.27±0.93 | 39.66±1.76 | 66.74±1.25 | 59.78±1.84 | 74.56±1.94 |
| | | ARI | 5.60±0.89 | 21.97±1.77 | 18.63±1.70 | 34.29±0.80 | 60.26±0.94 | 56.17±1.68 | 67.43±0.71 |
| | | F1 | 23.84±0.84 | 35.14±1.96 | 32.83±1.11 | 31.91±1.01 | 71.03±1.59 | 72.95±1.56 | 71.80±1.41 |
| | w/ | ACC | 67.21±1.74 | 42.82±1.81 | 41.42±1.98 | 51.42±0.79 | 76.10±1.19 | 75.51±1.67 | 79.39±1.97 |
| | | NMI | 41.43±1.43 | 34.47±1.87 | 31.15±1.01 | 39.26±1.70 | 66.55±1.16 | 58.91±1.79 | 74.89±1.88 |
| | | ARI | 37.39±1.21 | 21.51±1.89 | 19.05±1.80 | 34.27±0.78 | 61.14±1.21 | 55.55±1.55 | 67.58±0.83 |
| | | F1 | 45.93±1.34 | 35.37±1.94 | 31.86±1.17 | 31.23±1.26 | 70.43±1.77 | 73.25±1.56 | 71.45±1.48 |
| Reddit | w/o | ACC | 9.06±1.74 | 32.20±1.66 | OOM | 65.39±0.94 | OOM | 73.70±1.08 | 72.90±0.84 |
| | | NMI | 11.55±0.91 | 46.76±1.10 | | 70.30±1.73 | | 80.67±1.13 | 76.79±1.13 |
| | | ARI | 3.06±1.75 | 17.26±1.36 | | 63.32±1.75 | | 74.53±1.01 | 70.39±1.05 |
| | | F1 | 6.78±0.72 | 19.05±1.17 | | 51.39±1.78 | | 56.11±1.10 | 66.93±1.86 |
| | w/ | ACC | 37.59±0.97 | 31.71±1.77 | OOM | 65.83±0.98 | OOM | 74.66±1.29 | 72.64±0.92 |
| | | NMI | 21.04±1.09 | 47.69±0.97 | | 69.42±1.81 | | 80.66±0.87 | 76.25±1.24 |
| | | ARI | 25.10±1.50 | 17.51±1.51 | | 63.08±2.02 | | 74.52±1.18 | 70.12±1.10 |
| | | F1 | 36.80±2.13 | 18.76±1.13 | | 50.67±1.48 | | 55.29±0.84 | 66.74±1.94 |
| ogbn-arXiv | w/o | ACC | 18.15±1.00 | 22.39±1.88 | OOM | 29.72±1.17 | OOM | 35.02±1.94 | 41.57±1.78 |
| | | NMI | 22.29±1.70 | 70.51±1.16 | | 37.47±1.99 | | 46.36±1.86 | 42.67±0.78 |
| | | ARI | 7.48±0.92 | 63.60±0.84 | | 25.78±1.05 | | 26.80±1.30 | 34.35±1.39 |
| | | F1 | 13.07±1.66 | 51.39±0.91 | | 21.74±1.57 | | 23.04±1.22 | 26.10±1.00 |
| | w/ | ACC | 44.59±1.63 | 23.07±1.72 | OOM | 29.25±1.37 | OOM | 34.55±1.85 | 41.72±1.64 |
| | | NMI | 36.90±1.95 | 69.91±0.86 | | 37.61±2.02 | | 46.39±2.05 | 42.39±0.92 |
| | | ARI | 30.41±2.37 | 63.83±0.92 | | 25.14±1.21 | | 27.13±1.51 | 34.78±1.51 |
| | | F1 | 30.31±1.56 | 51.10±1.14 | | 21.77±1.50 | | 23.61±1.14 | 26.49±1.12 |
| ogbn-products | w/o | ACC | 18.24±1.76 | 31.56±1.30 | OOM | 35.39±1.07 | OOM | 40.07±1.02 | 39.01±1.59 |
| | | NMI | 22.24±1.41 | 41.14±0.92 | | 46.50±0.90 | | 53.65±1.17 | 48.77±1.16 |
| | | ARI | 7.39±1.34 | 22.13±1.33 | | 19.91±1.75 | | 22.81±0.71 | 20.95±1.30 |
| | | F1 | 12.77±1.47 | 22.98±1.65 | | 21.57±1.90 | | 25.01±1.02 | 24.14±1.03 |
| | w/ | ACC | 36.07±1.30 | 31.41±1.08 | OOM | 35.41±1.17 | OOM | 39.68±0.94 | 39.30±1.63 |
| | | NMI | 31.45±0.59 | 41.22±0.77 | | 46.24±0.62 | | 53.78±1.10 | 48.55±1.19 |
| | | ARI | 24.50±1.61 | 21.26±1.06 | | 20.13±1.55 | | 22.52±0.98 | 20.84±1.33 |
| | | F1 | 24.81±1.46 | 23.45±1.70 | | 20.69±2.07 | | 24.25±0.99 | 24.45±1.08 |

# E. URLs of Used Datasets

This section gives the URLs of the used benchmark datasets in Table 5.

- Cora: https://docs.dgl.ai/#CoraGraphDataset

- CiteSeer: https://docs.dgl.ai/#dgl.data.CiteseerGraphDataset

- Amazon-Photo: https://docs.dgl.ai/#dgl.data.AmazonCoBuyPhotoDataset

- ogbn-arxiv: https://ogb.stanford.edu/docs/nodeprop/#ogbn-arxiv

- Reddit: https://docs.dgl.ai/#dgl.data.RedditDataset

- ogbn-products: https://ogb.stanford.edu/docs/nodeprop/#ogbn-products

*Table 8.* Clustering performance of various DGC methods using multi-view representations derived from propagation-based hops. The reported results are averaged over ten runs on six benchmark datasets. All experiments are conducted under a 60% missing rate. "OOM" indicates an out-of-memory failure on a 24GB RTX 3090 GPU.

| Dataset | Metric | K-means | DGI | MVGRL | ProGCL | AGC-DRR | CCGC | S³GC | AMGC | Dink-Net |
|---|---|---|---|---|---|---|---|---|---|---|
| **Cora** | ACC | $41.58 \pm 2.45$ | $59.49 \pm 2.39$ | $63.88 \pm 2.53$ | $47.92 \pm 1.36$ | $44.73 \pm 1.57$ | $39.57 \pm 3.18$ | $66.38 \pm 2.64$ | $68.05 \pm 1.87$ | $68.91 \pm 1.65$ |
| | NMI | $18.78 \pm 1.32$ | $44.90 \pm 1.69$ | $64.53 \pm 2.85$ | $40.35 \pm 1.43$ | $28.45 \pm 1.94$ | $24.19 \pm 1.36$ | $49.37 \pm 1.62$ | $52.57 \pm 1.48$ | $51.57 \pm 1.57$ |
| | ARI | $21.40 \pm 1.27$ | $48.16 \pm 1.26$ | $53.79 \pm 2.79$ | $41.82 \pm 1.71$ | $18.38 \pm 1.04$ | $25.48 \pm 1.77$ | $50.02 \pm 2.31$ | $46.75 \pm 1.68$ | $48.56 \pm 2.21$ |
| | F1 | $44.87 \pm 2.05$ | $58.47 \pm 1.44$ | $65.78 \pm 2.37$ | $48.47 \pm 1.53$ | $50.75 \pm 2.05$ | $41.22 \pm 2.30$ | $66.80 \pm 1.77$ | $62.90 \pm 2.57$ | $67.27 \pm 1.98$ |
| **CiteSeer** | ACC | $46.54 \pm 1.49$ | $56.93 \pm 1.67$ | $57.33 \pm 0.68$ | $59.57 \pm 0.96$ | $54.05 \pm 1.63$ | $51.88 \pm 2.21$ | $64.31 \pm 1.53$ | $63.41 \pm 2.70$ | $65.28 \pm 1.55$ |
| | NMI | $25.25 \pm 2.50$ | $35.87 \pm 1.49$ | $34.55 \pm 2.52$ | $35.65 \pm 1.55$ | $32.84 \pm 3.01$ | $36.69 \pm 0.77$ | $37.85 \pm 1.40$ | $35.80 \pm 1.31$ | $37.69 \pm 1.41$ |
| | ARI | $25.10 \pm 3.01$ | $40.16 \pm 1.73$ | $28.13 \pm 1.43$ | $33.61 \pm 1.48$ | $34.74 \pm 2.51$ | $31.93 \pm 0.67$ | $37.23 \pm 2.27$ | $37.67 \pm 2.57$ | $38.17 \pm 1.59$ |
| | F1 | $39.46 \pm 2.13$ | $55.65 \pm 1.44$ | $55.31 \pm 2.42$ | $53.67 \pm 2.23$ | $57.50 \pm 0.74$ | $42.30 \pm 0.70$ | $58.57 \pm 2.20$ | $58.21 \pm 2.59$ | $58.53 \pm 1.41$ |
| **Amazon-Photo** | ACC | $33.26 \pm 1.99$ | $39.53 \pm 2.67$ | $38.66 \pm 1.45$ | $38.03 \pm 2.34$ | $70.83 \pm 2.26$ | $55.22 \pm 1.14$ | $73.73 \pm 0.53$ | $72.85 \pm 1.71$ | $75.41 \pm 1.47$ |
| | NMI | $23.80 \pm 1.54$ | $40.94 \pm 1.61$ | $26.90 \pm 1.77$ | $40.19 \pm 2.62$ | $62.24 \pm 1.68$ | $47.77 \pm 1.95$ | $67.08 \pm 1.50$ | $69.00 \pm 2.88$ | $68.63 \pm 1.81$ |
| | ARI | $16.30 \pm 1.73$ | $22.10 \pm 2.26$ | $37.86 \pm 1.49$ | $37.86 \pm 1.49$ | $54.16 \pm 1.51$ | $42.05 \pm 1.73$ | $61.78 \pm 1.69$ | $63.31 \pm 2.65$ | $63.69 \pm 2.86$ |
| | F1 | $31.63 \pm 2.34$ | $34.45 \pm 1.83$ | $39.78 \pm 1.95$ | $33.69 \pm 0.80$ | $67.82 \pm 1.69$ | $54.09 \pm 1.42$ | $67.55 \pm 1.87$ | $69.00 \pm 1.86$ | $68.85 \pm 1.45$ |
| **Reddit** | ACC | $24.76 \pm 2.26$ | $21.58 \pm 2.03$ | OOM | $62.83 \pm 1.78$ | OOM | OOM | $67.54 \pm 2.18$ | OOM | $66.96 \pm 2.14$ |
| | NMI | $10.82 \pm 1.95$ | $30.32 \pm 1.39$ | | $64.46 \pm 1.40$ | | | $68.84 \pm 2.00$ | | $68.95 \pm 1.70$ |
| | ARI | $12.90 \pm 2.24$ | $21.87 \pm 1.02$ | | $61.07 \pm 2.17$ | | | $62.90 \pm 1.38$ | | $61.78 \pm 1.76$ |
| | F1 | $23.98 \pm 2.83$ | $21.14 \pm 2.39$ | | $50.96 \pm 2.13$ | | | $56.98 \pm 1.43$ | | $60.53 \pm 2.22$ |
| **ogbn-arXiv** | ACC | $30.62 \pm 2.51$ | $29.16 \pm 2.01$ | OOM | $26.80 \pm 1.18$ | OOM | OOM | $33.72 \pm 2.04$ | OOM | $34.37 \pm 2.14$ |
| | NMI | $23.38 \pm 2.81$ | $35.66 \pm 1.87$ | | $30.48 \pm 1.19$ | | | $37.16 \pm 1.34$ | | $38.78 \pm 1.24$ |
| | ARI | $17.96 \pm 2.25$ | $12.24 \pm 2.23$ | | $23.89 \pm 2.71$ | | | $24.90 \pm 1.44$ | | $26.68 \pm 1.86$ |
| | F1 | $15.94 \pm 2.05$ | $23.58 \pm 2.11$ | | $16.17 \pm 3.12$ | | | $21.59 \pm 1.49$ | | $22.99 \pm 1.77$ |
| **ogbn-products** | ACC | $21.55 \pm 1.72$ | $30.21 \pm 1.96$ | OOM | $28.42 \pm 2.70$ | OOM | OOM | $32.33 \pm 1.94$ | OOM | $32.17 \pm 0.94$ |
| | NMI | $22.72 \pm 1.09$ | $40.83 \pm 2.55$ | | $41.90 \pm 1.80$ | | | $41.80 \pm 2.49$ | | $42.71 \pm 0.75$ |
| | ARI | $12.45 \pm 1.47$ | $15.78 \pm 2.29$ | | $12.56 \pm 2.29$ | | | $18.64 \pm 1.49$ | | $18.05 \pm 1.11$ |
| | F1 | $12.03 \pm 2.60$ | $17.38 \pm 1.07$ | | $15.50 \pm 2.88$ | | | $19.14 \pm 1.60$ | | $19.67 \pm 1.69$ |

# F. PyTorch-Style Pseudocode

We provide the PyTorch-style pseudocode for our *CMV-ND* in Algorithm 2.

**Algorithm 2** PyTorch-style pseudocode for *CMV-ND*

```
1  # diff_k_hop_priority_queue: Finds differential k-hop neighbors using a
       priority queue.
2  # graph: adjacency list representation of the graph.
3  # node: target node.
4  # k: hop distance.
5
6  def diff_k_hop_priority_queue(graph, node, k):
7      visited = set() # Track visited nodes
8      pq = [] # Priority queue storing (hop count, node)
9      heappush(pq, (0, node)) # Push the starting node into the queue
10     diff_hop = set() # Set to store differential k-hop nodes
11
12     while pq:
13         hops, curr_node = heappop(pq) # Process nodes based on hop count
               priority
14         if hops > k:
15             break
16         if hops == k and curr_node != node: # Only record differential nodes
17             diff_hop.add(curr_node)
18         if hops < k: # Expand neighborhood
19             for neighbor in graph[curr_node]:
20                 if neighbor not in visited:
21                     visited.add(neighbor)
22                     heappush(pq, (hops + 1, neighbor))
23     return diff_hop
24
25  # compute_augmented_features: Computes differential hop mean features.
26  # graph: adjacency list representation of the graph.
27  # node_features: feature matrix of nodes.
28  # max_hop: maximum hop distance to consider.
29
30  def compute_augmented_features(graph, node_features, max_hop):
31      num_nodes, feature_dim = node_features.shape
32      augmented_features = torch.zeros((max_hop + 1, num_nodes, feature_dim)) #
            Initialize augmented feature tensor
33
34      # Step 1: Assign original features as 0-hop representations
35      augmented_features[0] = node_features
36
37      # Step 2: Compute differential hop features from 1-hop to max_hop
38      for node in range(num_nodes): # Iterate over all nodes
39        for k in range(1, max_hop + 1): # Iterate over each hop distance
40            # Get differential k-hop neighborhood
41            diff_hop = diff_k_hop_priority_queue(graph, node, k)
42            if len(diff_hop) > 0:
43                # Compute mean feature of neighboring nodes
44                neighbor_features = node_features[list(diff_hop)]
45                hop_mean = neighbor_features.mean(dim=0)
46            else:
47                # If the differential neighborhood is empty, use a zero vector
48                hop_mean = torch.zeros(feature_dim)
49            # Store hop_mean in the corresponding position
50            augmented_features[k, node] = hop_mean
51
52      return augmented_features
```

