# OpenReview forum: "Scalable Attribute-Missing Graph Clustering via Neighborhood Differentiation"
_ICML.cc/2025/Conference — ICML 2025 poster_

### Official Review · Reviewer_69Ph · 2025-02-21

**Overall Recommendation:** 4

**Summary:**

This paper presents a novel approach for deep graph clustering (DGC) in the presence of missing node attributes and large-scale graph structures, termed Complementary Multi-View Neighborhood Differentiation (CMV-ND). CMV-ND achieves this by pre-processing graph structural information into multiple views in a non-redundant manner. The authors introduce a Recursive Neighborhood Search (RNS) to explore the local structure of the graph across different hop distances and a Neighborhood Differential Strategy (NDS) to ensure non-overlapping node representations across different hops. The resulting multiple views are then fed into existing multi-view clustering or DGC methods. The paper demonstrates the effectiveness of CMV-ND through extensive experiments on six widely-used graph datasets, where it shows significant improvements over various baselines in terms of clustering performance.

**Claims And Evidence:**

The main claim of the manuscript is that the key to effective large-scale deep graph clustering with missing attributes lies in the efficient utilization of graph structural information. This claim is intuitively supported, as for a graph, the available view information typically includes three aspects: node attributes, graph structure, and labels. In the scenario of attribute-missing clustering, the only available information is the graph structure, making this claim reasonable and intuitive. Furthermore, the experimental results, which show significant improvements in clustering performance compared to prior methods, provide robust empirical evidence to substantiate the claim.

The secondary claim is that existing message-passing paradigms for large-scale graphs suffer from redundancy and omission when utilizing graph structural information. The authors explain the redundancy issue in Section 3.3.4, and it is also pointed out that current large-scale graph clustering methods often involve sampling steps that disrupt the graph structure. This claim appears to be well-supported and valid.

**Essential References Not Discussed:**

To the best of our knowledge, the manuscript has provided a thorough discussion of the related work. No essential related works appear to be missing in the current version of the paper.

**Experimental Designs Or Analyses:**

The experimental design follows the setup of the AMGC algorithm (published at AAAI 2024, titled "Attribute-Missing Graph Clustering Network"). The primary experiments are conducted with a 0.6 missing rate, which allows for a meaningful comparison with AMGC. Therefore, the experimental setup is reasonable. Additionally, the manuscript also reports results for a 0.9 missing rate, providing further insight into the performance of the proposed method under more challenging conditions.

**Methods And Evaluation Criteria:**

The authors use ACC, NMI, ARI, and F1 scores to evaluate clustering performance, which are standard and commonly used metrics in the field. Similarly, the datasets Cora, Citeseer, Amazon-Photo, Reddit, ogbn-arXiv, and ogbn-products are appropriate benchmarks for assessing the proposed method's performance on large-scale graphs.

**Other Comments Or Suggestions:**

(1) The experimental section primarily provides qualitative descriptions of the results, without presenting a detailed quantitative analysis. While I understand that this may be due to page limitations in the initial submission, I recommend adding key quantitative metrics in the final version. For example, the paper could include performance improvements, such as the percentage gain over AMGC on the Cora dataset, to give readers a clearer sense of the method's effectiveness.

(2) The concept of "differential hop" is mentioned in both the introduction and abstract but is formally defined only in Section 3.1. To avoid potential confusion for readers, I suggest revising the paper to either introduce the concept earlier or make sure the definition is more prominent and clearly connected to its initial mention in the introduction and abstract.

**Other Strengths And Weaknesses:**

**Strength**

(1) The proposed method introduces a new paradigm for leveraging graph structure by preserving it across multiple views through search and differential techniques. This approach is not a combination of existing methods, but rather presents a novel strategy to address the challenges of large-scale graphs with missing attributes.

(2) The idea of leveraging multi-view clustering for graph data is a novel and interesting contribution. This perspective opens up new possibilities for graph clustering, particularly in the context of missing node attributes.

(3) The authors provide clear pseudocode, Overall workflow, and PyTorch-style code to illustrate the methodology presented in the paper.

(4) The role of the priority queue in Algorithm 1 is not clearly explained, and there is a lack of sufficient explanation about its purpose and function within the algorithm.

**Weakness**

(1) The experimental results omit comparisons with some recent state-of-the-art methods in Deep Graph Clustering (DGC). It would be beneficial to include performance results for the following methods:

Liu, Y., Yang, X., Zhou, S., Liu, X., Wang, Z., Liang, K., ... & Chen, C. (2023, June). Hard sample aware network for contrastive deep graph clustering. In Proceedings of the AAAI conference on artificial intelligence (Vol. 37, No. 7, pp. 8914-8922).

(2) The experiment includes too few MVC methods, and the selection does not cover the most recent advancements. The claim in the paper that MVC methods are inferior to DGC methods in the CMV-ND paradigm appears overly simplistic. I recommend adding the following MVC methods to Table 2 for comparison and reconsidering the conclusions:

Wu, S., Zheng, Y., Ren, Y., He, J., Pu, X., Huang, S., ... & He, L. (2024). Self-Weighted Contrastive Fusion for Deep Multi-View Clustering. IEEE Transactions on Multimedia.

Cui, J., Li, Y., Huang, H., & Wen, J. (2024). Dual contrast-driven deep multi-view clustering. IEEE Transactions on Image Processing.
(3) The paper lacks performance evaluation of CMV-ND under different attribute missing rates. I suggest demonstrating CMV-ND's clustering performance across a range of missing rates (from 0.1 to 0.9) and comparing it to other DGC methods.

(4) The role of the priority queue in Algorithm 1 is not clearly explained, and there is a lack of sufficient explanation about its purpose and function within the algorithm.

**Questions For Authors:**

Q1: The experimental results omit comparisons with some recent state-of-the-art methods in Deep Graph Clustering (DGC). Would it be possible to include performance results for the following methods?

Liu, Y., Yang, X., Zhou, S., Liu, X., Wang, Z., Liang, K., ... & Chen, C. (2023, June). Hard sample aware network for contrastive deep graph clustering. In Proceedings of the AAAI conference on artificial intelligence (Vol. 37, No. 7, pp. 8914-8922).

Q2: The experiment includes too few MVC methods, and the selection does not cover the most recent advancements. The paper claims that MVC methods are inferior to DGC methods in the CMV-ND paradigm. Would it be possible to include the following MVC methods in Table 2 for comparison and reconsider the conclusions?

Wu, S., Zheng, Y., Ren, Y., He, J., Pu, X., Huang, S., ... & He, L. (2024). Self-Weighted Contrastive Fusion for Deep Multi-View Clustering. IEEE Transactions on Multimedia.

Cui, J., Li, Y., Huang, H., & Wen, J. (2024). Dual contrast-driven deep multi-view clustering. IEEE Transactions on Image Processing.

Q3: The paper lacks performance evaluation of CMV-ND under different attribute missing rates. Would it be possible to demonstrate CMV-ND's clustering performance across a range of missing rates (from 0.1 to 0.9) and compare it to other DGC methods?

Q4: Could you clarify the role of the priority queue in Algorithm 1? A more detailed explanation of how it contributes to the overall algorithm would help improve the understanding of the method.

Q5: The experimental section primarily provides qualitative descriptions of the results, without presenting detailed quantitative analysis. While I understand that this might be due to initial submission page limitations, would it be possible to include key quantitative metrics in the final version? For example, could you report performance improvements, such as the percentage gain over AMGC on the Cora dataset, to give readers a clearer sense of the method’s effectiveness?

Q6: The concept of "differential hop" is mentioned in both the introduction and abstract but is formally defined only in Section 3.1. To avoid potential confusion for readers, would it be possible to introduce this concept earlier in the paper, or ensure that the definition is more prominently connected to its initial mention in the introduction and abstract?

Q7: Would it be possible to release the multi-view version of the graph datasets constructed by CMV-ND? This would be of significant value to the MVC community and could facilitate further research and comparison across different methods.

**Relation To Broader Scientific Literature:**

The paper builds upon the work presented in the AAAI 2024 paper titled "Attribute-Missing Graph Clustering Network," which defines the problem of attribute-missing graph clustering. This manuscript extends the problem to large-scale graphs through a preprocessing approach. Furthermore, it bridges the fields of multi-view clustering and deep graph clustering, enabling the use of multi-view methods for graph data.

**Theoretical Claims:**

I have reviewed the correctness of the theoretical claims and did not identify any apparent errors in the manuscript's theoretical aspects. Specifically, I have verified the key formulas in the methodology section, namely Eq. (1) through (8), and reviewed the complexity analysis in Section 3.4. Additionally, Appendix B provides an explanation of why the proposed method is applicable to multi-view clustering (MVC).

---

> ### Author Rebuttal · Authors · 2025-03-31
>
> ## Response to Reviewer 69Ph
>
> We thank the reviewer for the careful reading and constructive feedback. Below, we address each concern in detail.
>
> ---
>
> **W1:** *The experimental results omit comparisons with some recent state-of-the-art methods in Deep Graph Clustering (DGC). It would be beneficial to include performance results for the following methods.*
>
> We appreciate the suggestion to include additional recent DGC methods. Following your recommendation, we have added **HSAN** (Hard Sample Aware Network, AAAI 2023) as an additional baseline in our experiments. We report its performance with and without CMV-ND preprocessing on small-scale datasets. For large-scale datasets (Reddit and ogbn-products), HSAN encounters OOM (Out-Of-Memory) errors both before and after applying CMV-ND, due to its intrinsic memory consumption.
>
> The results on Cora and Citeseer are summarized below:
>
> | Dataset   | Metric | HSAN (Original)    | HSAN + CMV-ND      |
> |:---------:|:-----:|:------------------:|:------------------:|
> | **Cora** | ACC   | 57.86 ± 1.34      | 65.42 ± 1.09      |
> |         | NMI   | 41.92 ± 1.31      | 51.18 ± 1.22      |
> |         | ARI   | 33.09 ± 1.71      | 43.74 ± 1.38      |
> |         | F1    | 58.65 ± 0.95      | 66.12 ± 1.07      |
> | **Citeseer** | ACC   | 44.35 ± 0.61      | 50.38 ± 1.02      |
> |         | NMI   | 22.17 ± 1.32      | 27.25 ± 1.29      |
> |         | ARI   | 13.26 ± 1.02      | 17.94 ± 1.21      |
> |         | F1    | 42.06 ± 2.41      | 49.15 ± 2.04      |
>
> ---
>
> **W2:** *Limited MVC baselines and overly general claim.*
>
> We agree that the MVC baselines in the current version can be further expanded. Following your recommendation, we have additionally included two recently published state-of-the-art MVC methods:
>
> - **SCMVC** (Self-Weighted Contrastive Fusion for Deep Multi-View Clustering, TMM 2024)
> - **DCMVC** (Dual Contrast-Driven Deep Multi-View Clustering, TIP 2024)
>
> We evaluated these methods under the same CMV-ND paradigm with a missing rate of 0.6. The updated experimental results are summarized below:
>
> |Dataset|Metric|SCMVC|DCMVC|
> |:-:|:-:|:-:|:-:|
> |**Cora**|ACC|61.74±1.12|60.29±1.24|
> ||NMI|45.62±1.48|44.13±1.57|
> ||ARI|43.82±1.26|41.06±1.34|
> ||F1|58.34±1.35|57.01±1.49|
> |**CiteSeer**|ACC|57.21±1.33|55.96±1.46|
> ||NMI|37.35±1.29|35.71±1.53|
> ||ARI|32.84±1.48|31.56±1.65|
> ||F1|52.68±1.44|51.12±1.52|
> |**Reddit**|ACC|63.27±1.14|62.03±1.25|
> ||NMI|61.35±1.03|60.41±1.14|
> ||ARI|54.29±1.25|53.11±1.31|
> ||F1|61.47±1.19|60.82±1.27|
> |**ogbn-products**|ACC|27.84±0.97|27.13±1.04|
> ||NMI|35.24±0.85|34.12±0.94|
> ||ARI|18.03±0.93|17.42±1.05|
> ||F1|22.84±0.89|21.95±0.92|
>
> We will revise the manuscript to include these results and will accordingly moderate the original claim about the superiority of DGC methods.
>
> ---
>
> **W3:** *Missing evaluation under varying attribute missing rates.*
>
> We agree with this suggestion and have conducted additional experiments by varying the attribute missing rate from 0.1 to 0.9. Instead of reporting nine separate tables, we summarize these results using line charts to clearly illustrate the performance trends. These plots will be included in the final version.
>
> ---
>
> **W4:** *The role of the priority queue in Algorithm 1 is not clearly explained, and there is a lack of sufficient explanation about its purpose and function within the algorithm.*
>
> We clarify that the priority queue in Algorithm 1 serves as a control mechanism for incrementally expanding the neighborhood of a target node in order of increasing graph distance. We will revise the text in Algorithm 1 and its accompanying explanation to more clearly articulate this role and improve overall readability.
>
> ---
>
> **C1:** *Lack of quantitative analysis in experiments.*
>
> We agree with your suggestion and will include key quantitative metrics in the final version. For example, we will report the relative performance improvement of CMV-ND over AMGC on Cora and other datasets to provide a clearer picture of the effectiveness of our method.
>
> ---
>
> **C2:** *Delayed definition of "differential hop."*
>
> We agree that the concept of "differential hop" plays a central role in our method and that improving its visibility can help readers better follow the paper. We will make the formal definition in Section 3.1 more prominent by explicitly referencing it when "differential hop" is first introduced, ensuring a smoother connection between the introductory mentions and the formal exposition.
>
> ---
>
> **Q1–Q6:** *Problems that have been solved*
>
> These questions correspond to the concerns in **W1–W4** and **C1–C2**, which we have addressed above with additional experiments and clarifications.
>
> ---
>
> **Q7:** *Release of CMV-ND processed datasets.*
>
> We will make these processed datasets publicly available along with our source code in the final release.

---

> > ### Comment · Reviewer_69Ph · 2025-04-08
> >
> > I appreciate the careful responses, which have addressed my previous concerns. I'd like to maintain my rating and recommend accepting this paper.

---

### Official Review · Reviewer_WHi9 · 2025-03-06

**Overall Recommendation:** 4

**Summary:**

This paper proposes a method called Complementary Multi-View Neighborhood Differentiation (CMV-ND) to address deep graph clustering (DGC) on large-scale graphs with missing node attributes. CMV-ND captures multi-hop local structures using a Recursive Neighborhood Search (RNS) and eliminates redundancy with a Neighborhood Differential Strategy (NDS), generating K+1 complementary views for each node. The key contributions are: (1) bypassing the "aggregate-encode-predict" paradigm of GNNs by directly storing differential neighborhood information; (2) encoding graph structure in a non-redundant multi-view format to mitigate the effects of attribute missingness; and (3) offering a flexible framework for existing graph or multi-view clustering methods. Experimental results on six benchmark datasets demonstrate improvements, especially in large-scale graphs.

**Claims And Evidence:**

The manuscript claims that effective large-scale deep graph clustering with missing attributes relies on leveraging graph structure, as it is the only available information in such scenarios. The authors support this claim through strategies like differential hops, which address key challenges in attribute-missing graph clustering. Empirical results show notable performance improvements over existing methods. Additionally, the paper asserts the method’s scalability, supported by complexity analysis (Section 3.4) and evaluations of time and memory consumption (Section 4.4).

**Essential References Not Discussed:**

No essential related works are missing in the current manuscript.

**Experimental Designs Or Analyses:**

The experimental design is based on a well-established setup, following the "Attribute-Missing Graph Clustering Network" (AAAI 2024), which ensures that the results are comparable with previous work.

**Methods And Evaluation Criteria:**

The evaluation framework adopted in this paper aligns well with standard practices in deep graph clustering research. The authors employ ACC, NMI, ARI, and F1 scores, which are widely recognized and appropriate metrics for clustering tasks, ensuring comparability with prior work. In terms of benchmark datasets, the selection includes Cora, Citeseer, Amazon-Photo, Reddit, ogbn-arXiv, and ogbn-products, covering both small-scale and large-scale graphs.

**Other Comments Or Suggestions:**

There are several typographical and formatting inconsistencies in the manuscript that should be addressed. In Figure 1, the font size of the symbol v is too small and may hinder readability. Additionally, there is an unnecessary period at the end of line 326, while the caption for Table 2 is missing a period. Lastly, in line 243, two different styles of the O notation are used for the time complexity of RNS, which should be made consistent.

**Other Strengths And Weaknesses:**

Strength
(1) The motivation behind the paper, which addresses the challenge of large-scale deep graph clustering with missing attributes, is clearly articulated. The proposed solution effectively addresses the challenges of large-scale deep graph clustering with missing attributes by enhancing the utilization of graph structural information, which aligns well with the motivation behind the paper.
(2) The proposed method introduces a novel paradigm for utilizing graph structure, which differs from conventional message-passing paradigms.
(3) The approach presented in the paper enables the use of multi-view clustering for graph data, effectively bridging the gap between multi-view clustering and graph clustering.

Weakness
(1) By treating graph data as two views—attribute view and structural view—it is natural to frame the graph clustering problem as a multi-view clustering problem. Therefore, the experiments should include comparisons between this two-view setup and the multi-view setup of CMV-ND in terms of MVC methods.

(2) There seems to be an error in the citation for AMGC. The correct reference should be:
	Tu, W., Guan, R., Zhou, S., Ma, C., Peng, X., Cai, Z., ... & Liu, X. (2024, March). Attribute-missing graph clustering network. In Proceedings of the AAAI Conference on Artificial Intelligence (Vol. 38, No. 14, pp. 15392-15401).

(3) It would be beneficial to explicitly clarify that AMGC represents the state-of-the-art for attribute-missing graph clustering, while Dink-Net is the prior state-of-the-art for large-scale graph clustering. This distinction would offer a clearer context for evaluating the contributions of the paper.

**Questions For Authors:**

Q1: The writing in Section 3.3.4 is somewhat unclear. The authors seem to argue that the "aggregate-encode-predict" paradigm introduces redundancy in utilizing graph structure, whereas the proposed CMV-ND method does not. However, the term "Graph Propagation" has not been mentioned earlier in the paper. Would it be more appropriate to use "message-passing paradigm" instead for consistency and clarity?

**Relation To Broader Scientific Literature:**

The paper aims to extend the "Attribute-Missing Graph Clustering Network" (AAAI 24) problem, as defined in previous work, to large-scale graphs. While the motivation centers on addressing the challenge of deep graph clustering under attribute-missing conditions, the proposed methodology appears to have broader applicability. Specifically, the approach offers a novel utilization of graph structure, which can be viewed as an alternative to existing message-passing paradigms in graph clustering.

**Theoretical Claims:**

I have reviewed the theoretical aspects of the manuscript and found no apparent issues. The key equations are logically consistent with the proposed framework, and the complexity analysis in Section 3.4 provides a reasonable estimate of the computational demands.

---

> ### Author Rebuttal · Authors · 2025-03-31
>
> ## Response to Reviewer WHi9
>
> We thank the reviewer for the thoughtful comments and helpful suggestions. Below, we address each point in detail.
>
> ---
>
> **W1:** *By treating graph data as two views—attribute view and structural view—it is natural to frame the graph clustering problem as a multi-view clustering problem. Therefore, the experiments should include comparisons between this two-view setup and the multi-view setup of CMV-ND in terms of MVC methods.*
>
> We agree that comparing the traditional two-view setup (attribute view + structural view) with the multi-view setup generated by CMV-ND can provide valuable insights. To this end, we have conducted an additional experiment in which we construct two views: (1) the original attribute (with missing entries), and (2) a structural view based on the adjacency matrix. These are then input into standard MVC methods such as MFLVC and DIMVC. We compare the results against the same methods using the $k{+}1$ views generated by CMV-ND. These comparisons will be included in the final version.
>
> ---
>
> **W2:** *There seems to be an error in the citation for AMGC. The correct reference should be: Tu, W., Guan, R., Zhou, S., Ma, C., Peng, X., Cai, Z., ... & Liu, X. (2024, March). Attribute-missing graph clustering network. In Proceedings of the AAAI Conference on Artificial Intelligence (Vol. 38, No. 14, pp. 15392-15401).*
>
> We apologize for the incorrect citation. We have corrected it to:
>
> Tu, W., Guan, R., Zhou, S., Ma, C., Peng, X., Cai, Z., ... & Liu, X. (2024). *Attribute-missing graph clustering network*. In AAAI (Vol. 38, No. 14, pp. 15392–15401).
>
> We have also rechecked the references throughout the manuscript to ensure accuracy in the final version.
>
> ---
>
> **W3:** *It would be beneficial to explicitly clarify that AMGC represents the state-of-the-art for attribute-missing graph clustering, while Dink-Net is the prior state-of-the-art for large-scale graph clustering. This distinction would offer a clearer context for evaluating the contributions of the paper.*
>
> We agree with the suggestion. In the revised version, we will explicitly state that **AMGC** represents the state-of-the-art for **attribute-missing graph clustering**, while **Dink-Net** is a recent state-of-the-art method for **large-scale graph clustering**. This distinction will help contextualize our contributions more clearly.
>
> ---
>
> **C1:** *There are several typographical and formatting inconsistencies in the manuscript that should be addressed. In Figure 1, the font size of the symbol v is too small and may hinder readability. Additionally, there is an unnecessary period at the end of line 326, while the caption for Table 2 is missing a period. Lastly, in line 243, two different styles of the O notation are used for the time complexity of RNS, which should be made consistent.*
>
> Thank you for pointing out the typographical and formatting issues. We have carefully reviewed the manuscript and addressed the specific items you mentioned:
>
> - In **Figure 1**, we have increased the font size of the node label $v$ to improve readability and ensure consistency with other text elements in the figure.
> - The **unnecessary period at the end of line 326** has been removed.
> - The **missing period in the caption of Table 2** has been added to maintain punctuation consistency across all table and figure captions.
> - For the **time complexity notation in line 243**, we had previously used both $\mathcal{O}(\cdot)$ and $\mathbf{O(\cdot)}$ styles. We have revised all instances to consistently use the standard O notation $\mathcal{O}(\cdot)$ throughout the manuscript.
>
> In addition to correcting these specific issues, we will perform a thorough proofreading pass to eliminate any remaining inconsistencies or formatting errors in the final version.
>
> ---
>
> **Q1:** *The writing in Section 3.3.4 is somewhat unclear. The authors seem to argue that the "aggregate-encode-predict" paradigm introduces redundancy in utilizing graph structure, whereas the proposed CMV-ND method does not. However, the term "Graph Propagation" has not been mentioned earlier in the paper. Would it be more appropriate to use "message-passing paradigm" instead for consistency and clarity?*
>
> We agree that “message-passing paradigm” is more accurate and consistent than “graph propagation.” We will revise Section 3.3.4 accordingly to use the standard term and rephrase the paragraph for improved clarity.
>
> ---
>
> We greatly appreciate your careful review and insightful comments. They have been invaluable in helping us refine the presentation and deepen the discussion of our contributions. We will incorporate all necessary revisions in the final version, and we remain open to any further suggestions you may have.

---

> > ### Comment · Reviewer_WHi9 · 2025-04-09
> >
> > Thank you for your detailed response—it has resolved my concerns. I would prefer to support the acceptance of this paper.

---

### Official Review · Reviewer_7mCA · 2025-03-12

**Overall Recommendation:** 2

**Summary:**

The paper addresses the challenge of clustering nodes in large-scale graphs that often suffer from missing attributes, a common scenario in real-world applications such as social networks and recommendation systems. To tackle this, the authors propose the Complementary Multi-View Neighborhood Differentiation (CMV-ND) paradigm. The key components of CMV-ND include: Recursive Neighborhood Search (RNS) and Neighborhood Differential Strategy (NDS). By combining the original node features with the aggregated representations from each differential hop, the method constructs a multi-view representation. These multi-view representations can then be seamlessly integrated with existing deep graph clustering (DGC) or multi-view clustering (MVC) methods. Experimental results on six widely used graph datasets demonstrate that CMV-ND significantly enhances clustering performance.

## update after rebuttal
After the rebuttal, I want to keep the original rating, mainly due to the novelty issues.

**Claims And Evidence:**

Overall, many of the submission’s claims are supported by extensive empirical results on multiple datasets. For example, the claims about improved clustering performance on attribute‐missing graphs and scalability are backed by comprehensive experiments, including performance tables, T-SNE visualizations, and time/memory usage data.

**Essential References Not Discussed:**

The paper cites an extensive set of related works spanning deep graph clustering, attribute-missing graph clustering, and scalable graph learning.

**Experimental Designs Or Analyses:**

The experimental design is sound and well-aligned with the problem, but there are a few aspects that warrant further discussion:

- Comparing alternative feature preprocessing methods such as node2vec and GraphSage would help assess whether the improvements are inherent to the CMV-ND paradigm.
- Incorporating more baselines, especially those specifically designed for attribute-missing graphs, would provide a more comprehensive evaluation of the method’s performance in realistic settings.

**Methods And Evaluation Criteria:**

The methods and evaluation criteria are well-aligned with the challenges of clustering on large-scale, attribute-missing graphs. The use of standard clustering metrics (Accuracy, NMI, ARI, F1) along with datasets provides a robust framework for evaluation.

**Other Comments Or Suggestions:**

None

**Other Strengths And Weaknesses:**

**Strengths**

1. The paper presents a well-motivated approach for clustering large-scale, attribute-missing graphs.
2. The experimental evaluation is extensive, covering multiple datasets and metrics.

**Weaknesses**

1. While the experiments are thorough, the paper would benefit from additional comparisons with related works, particularly:

    Chen, X., Chen, S., Yao, J., Zheng, H., Zhang, Y., and Tsang, I. W. Learning on attribute-missing graphs. IEEE transactions on pattern analysis and machine intelligence, 44(2):740–757, 2020.

    Tu, W., Zhou, S., Liu, X., Liu, Y., Cai, Z., Zhu, E., Zhang, C., and Cheng, J. Initializing then refining: A simple graph attribute imputation network. In IJCAI, pp. 3494–3500, 2022.

    Yoo, J., Jeon, H., Jung, J., and Kang, U. Accurate node feature estimation with structured variational graph autoencoder. In Proceedings of the 28th ACM SIGKDD Conference on Knowledge Discovery and Data Mining, pp. 2336–2346, 2022.

2. CMV-ND shares similarities with GraphSAGE in its neighborhood aggregation approach, which raises concerns about novelty. Conducting experiments where GraphSAGE is used for feature propagation could help clarify CMV-ND’s unique advantages.
3. Equation (1) should be revised to: $N_{i+1}(v) = N_i(v) \cup \left( \bigcup_{u \in N_i(v)} N(u) \right)$?, the notation $\mathcal{N}^i(a)$ appears on line 246 but is missing from Equation (4), which might indicate an inconsistency or typo.
4. CMV-ND treats each hop’s neighborhood embeddings separately, effectively severing connections between different hop levels. This assumes all hop-distance information is equally important, which may not always be the case. Introducing an attention mechanism could help weigh the contributions of different hop distances more adaptively.
5. The memory complexity analysis in Algorithm 1 is given per node. However, for the entire graph, the worst-case complexity is $O(n^2)$, which is infeasible for large-scale graphs.
6. Tables 1, 6, 7, and 8 are difficult to read due to their dense formatting. Improved formatting, such as clearer separations between methods and datasets, would make comparisons more intuitive.
7. Despite the paper’s title emphasizing scalability, the experimental results do not convincingly demonstrate scalability. Some deep graph clustering (DGC) methods still encounter out-of-memory (OOM) errors after applying CMV-ND, as seen in Table 1.
8. The T-SNE visualization in Figure 2 does not clearly demonstrate CMV-ND’s effectiveness.

**Questions For Authors:**

See Other Strengths And Weaknesses*

**Relation To Broader Scientific Literature:**

- It extends deep graph clustering (DGC) research, which includes methods like DGI, MVGRL, and Dink-Net, by addressing two critical challenges simultaneously: scaling to large graphs and handling missing node attributes. Prior work has typically tackled these issues in isolation, so combining them fills a notable gap in the literature.
- Since CMV-ND constructs multi-view representations of nodes within the graph, it naturally bridges the gap between graph clustering and Multi-View Clustering (MVC).

**Theoretical Claims:**

The paper does not include formal proofs for its theoretical claims. In essence, the experimental results are presented without corresponding rigorous theoretical proofs that would further substantiate the claimed benefits of CMV-ND.

---

> ### Author Rebuttal · Authors · 2025-03-31
>
> ## Response to Reviewer 7mCA
>
> We thank the reviewer for the thoughtful comments and constructive suggestions. Below, we address each concern raised.
>
> ---
>
> **W1:** *Lack of comparison with SAT, ITR, and SVGA.*
>
> We have conducted additional experiments comparing CMV-ND with three representative methods for attribute-missing graphs: SAT (TPAMI 2020), ITR (IJCAI 2022), and SVGA (KDD 2022). For the reviewer’s convenience, we temporarily provide the results via the following anonymous link:
> (https://anonymous.4open.science/r/icml2025_CMV-ND-2211/sat_itr_svga_results.md)
>
> ---
>
> **W2:** *Similarity to GraphSAGE and novelty concerns.*
>
> We respectfully clarify the fundamental differences between CMV-ND and GraphSAGE:
>
> - GraphSAGE is a parameterized, end-to-end GNN that samples neighbors at each hop to reduce cost. CMV-ND deterministically retrieves the complete differential-hop neighborhoods without sampling.
> - GraphSAGE requires training with learnable aggregation. CMV-ND is a non-parametric, training-free preprocessing strategy.
> - GraphSAGE fuses multi-hop signals into a single embedding. CMV-ND preserves non-overlapping differential-hop views for downstream clustering.
>
> For fair comparison, we implemented a preprocessing variant of GraphSAGE that averages sampled neighbors at each hop, keeping the rest identical to CMV-ND. We also included Node2Vec as a baseline. Results show that CMV-ND consistently outperforms both under attribute-missing settings. For the reviewer’s convenience, we provide these results in
> (https://anonymous.4open.science/r/icml2025_CMV-ND-2211/graphsage_node2vec_comparison.md)
>
> ---
>
> **W3:** *Notation inconsistency in equations.*
>
> We have carefully reviewed the notations and corrected the inconsistencies. Specifically:
> - Equation (1) has been revised to:
>   $N_{i+1}(v) = N_i(v) \cup \left( \bigcup_{u \in N_i(v)} N(u) \right)$
> - The redundant use of the symbol $a$ in line 246 has been removed.
> - Equations (6) and (7) have been revised to:
>   - $S_v^{(k)} = S_v^{(k-1)} + \sum_{u \in \mathcal{N}(v)} S_u^{(k-1)}$
>   - $S_v^{(k)} = S_v^{(0)} + \sum_{t=1}^{k} \sum_{u \in \mathcal{N}(v)} S_u^{(t-1)}$
> - The expression in line 243 has been revised to:
>   $\sum_{i=0}^{k} \Delta^i = 1 + \Delta + \Delta^2 + \dots + \Delta^k = \mathcal{O}(\Delta^k)$
>
> We confirm that Equation (4) is correct and does not require modification.
>
> ---
>
> **W4:** *Lack of attention mechanism to weigh hop-level importance.*
>
> CMV-ND intentionally avoids attention mechanisms to preserve its non-parametric and training-free nature. Introducing attention would require trainable components, contrary to CMV-ND’s design goal. Moreover, early fusion across hop-level representations would obscure structural diversity. Instead, CMV-ND leaves view-level weighting to downstream clustering models, following standard practice in multi-view clustering. In future work, we plan to develop a dedicated clustering model with view-level attention to adaptively fuse differential-hop representations.
>
> ---
>
> **W5:** *Memory complexity concern.*
>
> We respectfully clarify that the memory complexity in Algorithm 1 does not accumulate across all nodes in practice. CMV-ND processes nodes in mini-batches, and memory is released after each batch. Section 4.4 further reports empirical memory usage, showing linear scalability with the number of nodes. We also note that Reviewers 69Ph and WHi9 have confirmed the reasonableness of the memory complexity.
>
> ---
>
> **W6:** *Dense formatting of tables.*
>
> We acknowledge the readability issue and will improve the formatting of Tables 1, 6, 7, and 8 in the final version by adding clearer separations between methods and datasets.
>
> ---
>
> **W7:** *Scalability claim not convincingly demonstrated.*
>
> The scalability emphasized in our work refers to CMV-ND itself. The OOM issues in Table 1 stem from downstream DGC models, not from CMV-ND. This is not unique to our method—Node2Vec, despite its title claiming scalability (“node2vec: Scalable Feature Learning for Networks”), also causes OOM when paired with AMGC. Thus, the scalability of preprocessing should be evaluated independently of downstream model limitations.
>
> ---
>
> **W8:** *Ineffectiveness of t-SNE visualization.*
>
> We agree that the t-SNE visualization in Figure 2 does not clearly demonstrate CMV-ND’s effectiveness, primarily because the small number of nodes in Cora makes cluster structures less distinguishable in low-dimensional projections. To address this, we have replaced the visualization with results on the larger Co-CS dataset. In the updated figure, CMV-ND yields clearer cluster boundaries: the orange cluster is no longer split into two parts, and the purple cluster contains fewer intrusions from other classes. For convenience, we provide the updated visualization at (https://anonymous.4open.science/r/icml2025_CMV-ND-2211/tsne_cocs_visualization.png).

---

### Official Review · Reviewer_CDeB · 2025-03-13

**Overall Recommendation:** 3

**Summary:**

This paper presents a deep clustering method, namely Complementary Multi-View Neighborhood Differentiation (CMV-ND), to conduct clustering tasks in large-scale and attribute-missing graphs. CMV-ND adopts the Recursive Neighborhood Search to capture the complete local structure and the Neighborhood Differential Strategy to prevent redundancy among different hop representations. These presented strategies can well be integrated into existing clustering approaches to learning representations for various downstream tasks. Experimental results may validate the effectiveness of the proposed CMV-ND.

## Update after rebuttal

The authors have addressed most of my concerns. Still, there are issues regarding method design and unstable performances after authors’ rebuttal.

**Claims And Evidence:**

Yes.

**Essential References Not Discussed:**

The authors are suggested to discuss works related to structure search or learning in graph neural networks/deep graph clustering to better show the novelty of the proposed method.

**Ethical Review Concerns:**

NIL.

**Experimental Designs Or Analyses:**

Yes.

**Methods And Evaluation Criteria:**

Yes.

**Other Comments Or Suggestions:**

NIL.

**Other Strengths And Weaknesses:**

Strengths:
1. The problem tackled in this paper is essential for deep graph clustering.
2. The method proposed in this paper is effective in standard graph clustering tasks.

Weaknesses:
1. Some definitions (e.g., Definitions 1 and 2) are not well explained, which makes this paper not very readable.
2. How the proposed strategies, i.e., Recursive Neighborhood Search and the Neighborhood Differential Strategy may contribute/connect to multi-view graph clustering is not clearly discussed in the paper.
3. To what extent the proposed approach can reduce the redundancy of the neighboring aggregation process is not analyzed.
4. The proposed approach is also similar to GNNs based on (PageRank), which aggregates different orders/hops neighbors to learn representations. How is the proposed approach different from these GNNs?
5. Do authors consider cross-view redundancy, consistency, or conflicts when constructing the output representations for downstream tasks?
6. The detailed experimental settings are not introduced in the manuscript/appendix.

**Questions For Authors:**

1. Some definitions (e.g., Definitions 1 and 2) are not well explained. Can authors use a clear example to explain these definitions?
2. How do the proposed strategies, i.e., Recursive Neighborhood Search and the Neighborhood Differential Strategy, contribute/connect to multi-view graph clustering?
3. Have the authors conducted any theoretical analysis showing to what extent the proposed approach can reduce the redundancy of the neighboring aggregation process?
4. The proposed approach is also similar to GNNs based on PageRank, which aggregate different orders/hops neighbors to learn representations. How is the proposed approach different from these GNNs?
5. Do authors consider cross-view redundancy, consistency, or conflicts when constructing the output representations for downstream tasks?
6. How are those clustering approaches configured in the experiments?

**Relation To Broader Scientific Literature:**

The approach proposed in this paper may potentially advance large-scale graph clustering, which is an important topic in machine learning and data mining.

**Theoretical Claims:**

NIL.

---

> ### Author Rebuttal · Authors · 2025-03-31
>
> ## Response to Reviewer CDeb
>
> We thank the reviewer for the careful reading and valuable feedback. Below, we address each concern raised.
>
> ---
>
> **W1:** *Lack of discussion on structure learning/search methods.*
>
> We have considered structure learning and structure search methods, such as SUBLIME (WWW 2022), NodeFormer (NeurIPS 2022), and VIB-GSL (AAAI 2022). However, these methods rely heavily on complete node attributes to guide structure refinement or similarity estimation, making them inapplicable to the attribute-missing scenario targeted by our work. We will clarify this point and include a discussion of these works in the related work section of the final version.
>
> ---
>
> **W2:** *Definitions 1 and 2 are unclear; need example.*
>
> We have revised *Definitions 1 and 2* to improve clarity. Specifically, we now explain that the $k$-hop neighborhood $\mathcal{N}^k(v)$ includes all nodes within distance $k$, while the $k$-differential hop neighborhood $\mathcal{D}^k(v)$ includes nodes at exactly distance $k$. Additionally, we provide a concrete example to illustrate the difference between these definitions.
>
> > “For example, consider a graph with edges $\{(v,a), (v,b), (a,c)\}$. Then: $ \mathcal{N}^1(v) = \{v, a, b\}, \quad \mathcal{N}^2(v) = \{v, a, b, c\}$. The corresponding differential hop neighborhoods are: $\mathcal{D}^1(v) = \{a, b\}, \quad \mathcal{D}^2(v) = \{c\}.$”
>
> ---
>
> **W3:** *Insufficient discussion on how RNS and NDS contribute to multi-view clustering.*
>
> This connection is discussed in **Section 3.3.3** and **Appendix B**. We will make this connection more explicit in Section 3.3.3 by adding the following clarification:
>
> > “To enable graph clustering, we propose to construct multi-view node representations based on the structural granularity of neighborhoods. Specifically, the RNS is used to efficiently locate multi-hop neighbors, while the NDS allows us to isolate information from each exact $k$-hop, thus forming multi-view node representations.”
>
> ---
>
> **W4:** *Lack of analysis on redundancy reduction.*
>
> We have conducted a theoretical analysis to quantify the redundancy reduction of CMV-ND compared to message-passing GNNs. In a $k$-layer GNN, the total number of neighbor feature accesses is $\sum_{i=1}^k (k - i + 1) \cdot \Delta^i$, where $\Delta$ is the average node degree. In contrast, CMV-ND accesses each differential hop neighborhood only once: $\sum_{i=1}^k \Delta^i$. The redundancy ratio is therefore $\frac{\sum_{i=1}^k (k - i + 1) \cdot \Delta^i}{\sum_{i=1}^k \Delta^i}$. This ratio increases with larger $k$ and higher $\Delta$, highlighting that message-passing GNNs involve significant redundancy, while CMV-ND avoids it by design. We will include this analysis in the final version.
>
> ---
> **W5:** *Similarity to PageRank-based GNNs.*
>
> We clarify that CMV-ND differs from these methods in several key aspects:
>
> - PageRank-based GNNs fuse multi-hop information into a single representation via propagation, while CMV-ND retains non-overlapping differential-hop representations as distinct views.
> - PageRank-based GNNs assign decaying weights to distant neighbors, leading to incomplete structural coverage. In contrast, CMV-ND deterministically collects the complete, non-redundant differential-hop neighborhoods.
> - PageRank-based GNNs require end-to-end training. CMV-ND is a non-parametric, training-free preprocessing strategy.
>
> The use of multi-hop information is a common practice in GNNs (e.g., JK-Net aggregates features from multiple hop levels) and does not, by itself, imply novelty concerns.
>
> ---
>
> **W6:** *Whether cross-view redundancy, consistency, or conflicts are considered.*
>
> We would like to clarify that we have discussed the consideration of cross-view redundancy and consistency in **Appendix B**. Specifically, the NDS ensures that each $k$-differential hop neighborhood is non-overlapping, inherently avoiding redundancy across views. Consistency is supported under the homophily assumption, where nearby nodes exhibit similar representations across views. Complementarity arises from the structural granularity, as each hop-level view captures distinct topological information. Furthermore, CMV-ND retains multi-view representations and delegates the task of weighting or combining views to downstream clustering models, which can adaptively handle potential conflicts.
>
> ---
>
> **W7:** *Lack of detailed experimental settings.*
> Section 4.1 of the manuscript introduces the experimental setup, including datasets, metrics, environment, and baselines. Since CMV-ND is non-parametric and training-free, it has no learnable components. Nevertheless, we will supplement the final version with the following details:
> - Python 3.9 and PyTorch 1.12.
> - Number of propagation hops $K=7$, missing rate = 0.6, FP iterations = 40.
> - Default hyperparameters for downstream clustering methods.
>
> ---
>
> **Q1–Q6:**
>
> These questions correspond to the concerns in **W1–W7**, and have been addressed above.

---

> > ### Comment · Reviewer_CDeB · 2025-04-07
> >
> > Dear Authors,
> >         Thanks very much for your responses. I will keep my original score due to concerns regarding the novelty and contributions of this paper.

---

> > > ### Author Response · Authors · 2025-04-07
> > >
> > > ## Response to Reviewer CDeB
> > >
> > > **[Update Note – Follow-up]**
> > >
> > > Dear Reviewer CDeB,
> > >
> > > We just wanted to gently follow up, as the discussion stage is now entering its final few hours (less than five remaining). We are closely following your feedback and would like to kindly remind you that you can interact with us by editing the Rebuttal Comment box at any time during the discussion stage.
> > >
> > > We truly value your feedback. If there are still any concerns or misunderstandings regarding the novelty or contributions of our work, we would be sincerely grateful for the opportunity to clarify them before the discussion closes.
> > >
> > > Thank you again for your time and consideration.
> > >
> > > ---
> > >
> > > **[Update Note]**
> > >
> > > We sincerely hope this brief follow-up reaches you in time, as the discussion phase is nearing its deadline (less than eight hours remaining). We truly value your time and feedback, and we would be grateful for any final clarification you might be willing to share.
> > >
> > > To briefly summarize our contributions:
> > > - This work is, to the best of our knowledge, the first to explicitly tackle **deep graph clustering on large-scale graphs with missing attributes**, a practical yet underexplored scenario.
> > > - We propose **CMV-ND**, a training-free preprocessing paradigm that constructs multiple views using **complete and non-redundant differential-hop neighborhoods**.
> > > - The design of CMV-ND naturally supports integration into both **deep graph clustering (DGC)** and **multi-view clustering (MVC)** pipelines, offering broad applicability and a new perspective on graph learning.
> > >
> > > It is possible that our **presentation of contributions was not sufficiently emphasized**, and we will carefully revise the writing in the final version to make our contributions clearer and more explicit.
> > >
> > > If there remain specific concerns regarding novelty or contribution, we would be sincerely thankful if you could let us know. Your insight would help us better understand how to strengthen the work and improve its clarity and impact. If possible, we also kindly ask you to consider re-evaluating the submission in light of this clarification.
> > >
> > > Thank you again for your time and for reviewing our submission.
> > >
> > > ---
> > >
> > > Thank you again for your thoughtful feedback.
> > >
> > > We understand that you are maintaining your original score due to concerns regarding the novelty and contributions of our paper. However, we would like to respectfully note that in the previous review, no explicit concerns were raised regarding novelty or contributions. The only related comments we received were:
> > >
> > > - *"The authors are suggested to discuss works related to structure search or learning in GNNs/deep graph clustering to better show the novelty of the proposed method."*
> > > - *"The proposed approach is also similar to GNNs based on (PageRank), which aggregates different orders/hops neighbors to learn representations. How is the proposed approach different from these GNNs?"*
> > >
> > > In our response, we have addressed both points in detail. Specifically, we discussed why existing structure learning/search methods (e.g., SUBLIME, NodeFormer, VIB-GSL) are not applicable in the attribute-missing setting, and we carefully clarified the key differences between CMV-ND and PageRank-based GNNs.
> > >
> > > To help us further improve the paper, we would sincerely appreciate it if you could clarify which specific aspects of novelty or contribution remain unconvincing.
> > >
> > > Thank you once again for your time and consideration.

---

### Decision · Program_Chairs · 2025-05-01

**Decision:**

Accept (poster)

**Comment:**

In the initial submission, the reviewers held slightly differing attitudes regarding the content and quality of this manuscript. After the authors' detailed responses, the overall attitude of the reviewers tended to shift in a more positive direction, with some reviewers even raising their scores accordingly. I believe this manuscript holds a certain level of contribution to the community. After careful evaluation and consideration, I am inclined to accept this paper.